# Macrophage-Myofibroblast Transition Contributes to Myofibroblast Formation in Proliferative Vitreoretinal Disorders

**DOI:** 10.3390/ijms241713510

**Published:** 2023-08-31

**Authors:** Ahmed M. Abu El-Asrar, Gert De Hertogh, Eef Allegaert, Mohd I. Nawaz, Sara Abouelasrar Salama, Priscilla W. Gikandi, Ghislain Opdenakker, Sofie Struyf

**Affiliations:** 1Department of Ophthalmology, College of Medicine, King Saud University, Riyadh 11411, Saudi Arabia; mnawaz@ksu.edu.sa (M.I.N.); pgikandi.c@ksu.edu.sa (P.W.G.); ghislain.opdenakker@kuleuven.be (G.O.); 2Dr. Nasser Al-Rashid Research Chair in Ophthalmology, College of Medicine, King Saud University, Riyadh 11411, Saudi Arabia; 3Laboratory of Histochemistry and Cytochemistry, University of Leuven, KU Leuven, 3000 Leuven, Belgium; gert.dehertogh@uzleuven.be (G.D.H.); eef.allegaert@kuleuven.be (E.A.); 4University Hospitals UZ Gasthuisberg, 3000 Leuven, Belgium; 5Laboratory of Molecular Immunology, Rega Institute for Medical Research, Department of Microbiology, Immunology and Transplantation, University of Leuven, KU Leuven, 3000 Leuven, Belgium; sarah.abuelasrar@kuleuven.be (S.A.S.); sofie.struyf@kuleuven.be (S.S.); 6Laboratory of Immunobiology, Rega Institute for Medical Research, Department of Microbiology, Immunology and Transplantation, University of Leuven, KU Leuven, 3000 Leuven, Belgium

**Keywords:** proliferative diabetic retinopathy, proliferative vitreoretinopathy, macrophage-myofibroblast transition, M2 macrophages

## Abstract

Inflammation and fibrosis are key features of proliferative vitreoretinal disorders. We aimed to define the macrophage phenotype and investigate the role of macrophage-myofibroblast transition (MMT) in the contribution to myofibroblast populations present in epiretinal membranes. Vitreous samples from proliferative diabetic retinopathy (PDR), proliferative vitreoretinopathy (PVR) and nondiabetic control patients, epiretinal fibrovascular membranes from PDR patients and fibrocellular membranes from PVR patients, human retinal Müller glial cells and human retinal microvascular endothelial cells (HRMECs) were studied by ELISA, immunohistochemistry and flow cytometry analysis. Myofibroblasts expressing α-SMA, fibroblast activation protein-α (FAP-α) and fibroblast-specific protein-1 (FSP-1) were present in all membranes. The majority of CD68^+^ monocytes/macrophages co-expressed the M2 macrophage marker CD206. In epiretinal membranes, cells undergoing MMT were identified by co-expression of the macrophage marker CD68 and myofibroblast markers α-SMA and FSP-1. Further analysis revealed that CD206^+^ M2 macrophages co-expressed α-SMA, FSP-1, FAP-α and ß-catenin. Soluble (s) CD206 and sFAP-α levels were significantly higher in vitreous samples from PDR and PVR patients than in nondiabetic control patients. The proinflammatory cytokine TNF-α and the hypoxia mimetic agent cobalt chloride induced upregulation of sFAP-α in culture media of Müller cells but not of HRMECs. The NF-ĸß inhibitor BAY11-7085 significantly attenuated TNF-α-induced upregulation of sFAP-α in Müller cells. Our findings suggest that the process of MMT might contribute to myofibroblast formation in epiretinal membranes, and this transition involved macrophages with a predominant M2 phenotype. In addition, sFAP-α as a vitreous biomarker may be derived from M2 macrophages transitioned to myofibroblasts and from Müller cells.

## 1. Introduction

Proliferative vitreoretinal disorders, comprising proliferative diabetic retinopathy (PDR) and proliferative vitreoretinopathy (PVR), are characterized by the pathological development of membranes on the surface of the retina. In the epiretinal membranes of patients with PDR and PVR, inflammatory and fibrotic changes are characterized by the presence of CD68^+^ monocytes/macrophages and α-smooth muscle actin (α-SMA)-expressing myofibroblasts [1,2,3,4,5] and contribute to the initiation and progression of proliferative vitreoretinal disorders.

Macrophages are considered to be a key cell type behind the development of fibrotic disease [6,7,8]. Macrophage phenotype, which is broadly stratified into classically activated or M1 and alternatively activated or M2, has been shown to influence pathologic fibrosis. M1 macrophages carry the cell surface marker CD86, a costimulatory molecule, and express overtly inducible nitric oxide synthase (iNOS). M2 macrophages have the cell surface markers CD206 (known as the macrophage mannose receptor) and CD163 (known as the hemoglobin-haptoglobin scavenger receptor). Specifically, the M2 macrophage phenotype has been identified as a driver of fibrosis by producing pro-fibrotic cytokines such as transforming growth factor-β1 (TGF-β1) [6,7,8]. In contrast, M1 macrophages lack expression of pro-fibrotic cytokines while they secrete extracellular matrix-degrading proteases, suggesting that M1 macrophages have an anti-fibrotic effect [8]. 

Myofibroblasts are the key cellular mediators of fibrosis, and their presence is a marker of progressive fibrosis. Myofibroblasts have contractile properties and generate pro-fibrotic mediators and extracellular matrix components, including collagen [9]. Identification of fibroblasts/myofibroblasts is based on the expression of various markers, such as α-SMA, fibroblast activation protein-α (FAP-α) and fibroblast-specific protein-1 (FSP-1, known as S100A4) [10]. The added value of FAP-α is that it marks fibroblast/myofibroblast activation in fibrotic conditions [11,12]. Their cellular origin in the ocular microenvironment of patients with PDR and PVR is not entirely clear. Accumulating evidence indicates a diverse cellular origin of these myofibroblasts, including bone marrow-derived fibrocytes and endothelial-to-mesenchymal transition (EndoMT) [3,4]. Epithelial-to-mesenchymal transition (EMT) of retinal pigment epithelial (RPE) cells (expressing cytokeratin) to a myofibroblast-like phenotype was suggested to be a major pathological process in PVR [13,14]. In addition, vitreal hyalocytes-to-myofibroblast transdifferentiation was suggested to have a role in the pathogenesis of PDR [15]. 

Macrophage-myofibroblast transition (MMT) is a term that describes the mechanism through which macrophages, derived from circulating monocytes originating in the bone marrow, transform into myofibroblasts and thereby contribute to fibrosis [16,17,18,19,20,21,22,23]. The transitioning cells express markers of both the macrophages and myofibroblast lineage. MMT was demonstrated to constitute an additional source of myofibroblasts in experimental models of fibrosis, such as lung fibrosis, renal fibrosis after transplantation or ureteric obstruction and post-myocardial infarction fibrosis [16,17,18,19,20,21,22]. Those findings were confirmed in patients with active chronic renal allograft rejection [17] and patients with fibrotic lung disease [22]. MMT was also identified as a mechanism for the tumor microenvironment to promote cancer-associated fibroblast formation [23]. The study of MMT in fibrotic diseases relies on the detection of intermediate cells that co-express macrophage markers, such as CD68 and myofibroblast markers, such as α-SMA [16,17,18,19,20,21,22,23]. Several studies demonstrated that the TGF-β1/β-catenin signaling pathway promotes the MMT process [6,7,8,16,17,21,22,23,24]. 

The four goals of the present study were (1) to define the macrophage population in epiretinal membranes from patients with PDR and PVR in the context of classically activated or M1 and alternatively activated or M2 macrophage phenotypes; (2) to reveal the presence of macrophages co-expressing myofibroblast markers as an indication of the occurrence of the MMT process contributing to the myofibroblast population present in the epiretinal membranes; (3) to analyze vitreous fluid from patients for the expression levels of the MMT-related markers: soluble (s)CD206 which is a marker of activated M2 macrophages and sFAP-α, which is a marker of myofibroblast activation, thereby evaluating their biomarker potential for proliferative vitreoretinal disorders; (4) to evaluate the expression of FAP-α by cultured human retinal Müller glial cells and human retinal microvascular endothelial cells (HRMECs) as a source of sFAP-α in the vitreous fluid, since these cells are the major cell types in inflammatory reactions in the retina. 

## 2. Results

### 2.1. Neovascularization and Expression of Myofibroblast Markers in Epiretinal Fibrovascular Membranes from Patients with PDR

No staining was detected in the negative control slides (Figure 1A). Neovessels that were positive for the vascular endothelial cell marker CD31 were detected in all 12 membranes (Figure 1B). Strong cytoplasmic immunoreactivity for α-SMA was noted in spindle-shaped myofibroblasts (Figure 1C). Membranous immunoreactivity for FAP-α was noted in spindle-shaped myofibroblast-like cells (Figure 1D). Immunoreactivity for FSP-1 was noted in vascular endothelial cells lining new blood vessels (Figure 1E) and stromal spindle-shaped cells (Figure 1F). Tissue pigmentation was absent in PDR epiretinal membranes (Figure 1).

### 2.2. Expression of Myofibroblast Markers in Epiretinal Fibrocellular Membranes from Patients with PVR

All twelve PVR membranes were characterized by the presence of heavy pigmentation (Figure 2). No staining was observed in the negative control slide (Figure 2A). Immunoreactivities for α-SMA (Figure 2B), FAP-α (Figure 2C) and FSP-1 (Figure 2D) were detected in spindle-shaped myofibroblast-like cells in all membranes.

### 2.3. CD68^+^ Monocytes/Macrophages Are Predominantly CD206^+^ M2 Phenotype in Epiretinal Membranes

Immunohistochemical analysis revealed the presence of CD68^+^ monocytes/macrophages in PDR epiretinal membranes (Figure 3A). Similarly, monocytes/macrophages expressing CD68 were detected in PVR epiretinal membranes (Figure 3B). Pigment was detected within some of the CD68^+^ cells in PVR epiretinal membranes (Figure 3B).

To confirm that M2 is the predominant phenotype of monocytes/macrophages in epiretinal membranes, we performed double-labeling experiments. Double immunohistochemistry analysis demonstrated that the majority of CD68^+^ monocytes/macrophages co-expressed CD206 in epiretinal membranes from patients with PDR (Figure 4A) and PVR (Figure 4B).

### 2.4. CD206^+^ Cells in Epiretinal Membranes from Patients with PDR and PVR

PDR epiretinal membranes contained CD206 immunoreactivity in vascular endothelial cells lining pathologic new blood vessels (Figure 5A). CD206 was also detected in stromal monocyte/macrophages (Figure 4A and Figure 5B) and spindle-shaped stromal cells (Figure 5C). In serial sections, the distribution and morphology of spindle-shaped cells expressing CD206 were similar to those of myofibroblasts expressing α-SMA (Figure 1C). CD206 expression by retinal endothelial cells was confirmed on in vitro primary HRMEC culture with the use of flow cytometry, wherein 29 ± 5.2% of cells were positive for CD206 expression (Figure 5D).

In PVR epiretinal membranes, immunoreactivity for CD206 was detected in monocytes/macrophages (Figure 4B and Figure 5E) and in spindle-shaped myofibroblast-like cells (Figure 5F). Some of the CD206^+^ monocytes/macrophages in PVR epiretinal membranes contained pigment (Figure 5E). Immunoreactivities for M1-related CD86 and iNOS were not detected in epiretinal membranes from patients with PDR and PVR.

### 2.5. Characterization of the Cells Containing Pigment in Epiretinal Fibrocellular Membranes from Patients with PVR

To characterize the heavily pigmented cells in PVR epiretinal membranes, we performed immunohistochemical stainings with the RPE marker pan-cytokeratin and with the macrophage markers CD68 and CD206 in serial sections. The heavily pigmented cells were negative for pan-cytokeratin (Figure 6A) but were positive for CD68 (Figure 6B) and CD206 (Figure 6C). These findings suggest that these cells are monocytes/macrophages engulfing pigments. In addition, we observed the presence of pan-cytokeratin expressing RPE cells in PVR epiretinal membranes (Figure 6A).

### 2.6. Intermediate CD68^+^/α-SMA^+^ and CD68^+^/FSP-1^+^ Double-Positive Cells in Epiretinal Membranes from Patients with PDR and PVR

Double-labeling experiments were performed to identify cells undergoing MMT on the basis of co-expression of macrophage and myofibroblast markers. On the one hand, double immunohistochemistry analysis demonstrated that most of the CD68^+^ monocytes/macrophages co-expressed α-SMA (Figure 7A) and FSP-1 (Figure 7B) in PDR epiretinal membranes. Similarly, most of the CD68^+^ monocytes/macrophages co-expressed α-SMA (Figure 7C) and FSP-1 (Figure 7D) in PVR epiretinal membranes. In epiretinal membranes from patients with PDR, 87% (range: 82–89%; n = 9 patients) and 88% (range: 83–93%; n = 5 patients) of CD68^+^ monocytes/macrophages co-expressed α-SMA and FSP-1, respectively. In epiretinal membranes from patients with PVR, CD68^+^/α-SMA^+^ cells accounted for 85% (range: 77–92%; n = 8 patients) of the total CD68+ monocyte/macrophage population and 97% (range: 95–98%; n = 3 patients) of the CD68^+^ monocytes/macrophages co-expressed FSP-1. On the other hand, double immunohistochemistry did not reveal the presence of CD68^+^/FAP-α^+^ cells in epiretinal membranes. These data suggest that MMT is contributing to the accumulation of myofibroblasts in epiretinal membranes.

### 2.7. CD206^+^ M2 Macrophages Contribute to MMT in Epiretinal Membranes from Patients with PDR and PVR

In PDR epiretinal membranes, double immunohistochemistry analysis revealed that the majority of CD206^+^ M2 monocytes/macrophages co-expressed α-SMA (Figure 8A), FSP-1 (Figure 8B) and FAP-α (Figure 8C). CD206^+^/α-SMA^+^ cells, CD206^+^/FSP-1^+^ cells and CD206^+^/FAP-α^+^ cells accounted for 81% (range: 75–86%; n = 9 patients), 83% (range: 78–92%; n = 7 patients) and 70% (range: 63–75%; n = 8 patients) of the total CD206^+^ M2 macrophage population. Similarly, CD206^+^/α-SMA^+^ (Figure 9A), CD206^+^/FSP-1^+^ (Figure 9B) and CD206^+^/FAP-α^+^ (Figure 9C) double-positive cells were identified in PVR epiretinal membranes. In PVR epiretinal membranes, all CD206^+^ M2 macrophages co-expressed α-SMA and FSP-1 (n = 4 patients and n = 3 patients, respectively). CD206^+^/FAP-α^+^ cells accounted for 85% (range: 80–90%; n = 5 patients) of the total CD206^+^ macrophage population. It is worth noting that most of the CD206^+^/FAP-α^+^ cells were spindle-shaped.

### 2.8. Expression of TGF-ß1 and ß-Catenin in Epiretinal Membranes from Patients with PDR and PVR

We next evaluated the expression of the components of the TGF-ß1/ß-catenin signaling pathway, as it is the main mediator of MMT. In PDR epiretinal membranes, immunoreactivity for TGF-ß1 was detected in vascular endothelial cells and stromal spindle-shaped myofibroblast-like cells (Figure 10A). In PVR epiretinal membranes, immunoreactivity for TGF-ß1 was observed in spindle-shaped myofibroblast-like cells (Figure 10B).

In PDR epiretinal membranes, ß-catenin expression was observed in endothelial cells lining new blood vessels (Figure 11A). In the stroma, ß-catenin expression was detected in spindle-shaped myofibroblast-like cells (Figure 11B) and monocytes/macrophages expressing CD206 (Figure 11C). In PVR epiretinal membranes, ß-catenin was expressed in spindle-shaped myofibroblast-like cells (Figure 12A) and CD206^+^ monocytes/macrophages (Figure 12B). Table 1 provides a summary of the immunohistochemical findings.

### 2.9. Levels of sCD206 and sFAP-α in Vitreous Samples from Control Patients and Patients with PDR and PVR

To provide an additional indication of the occurrence of MMT in the ocular microenvironment of patients with proliferative vitreoretinal disorders, we measured the levels of sCD206 and sFAP-α in vitreous samples from nondiabetic control patients with rhegmatogenous retinal detachment, PDR patients and PVR patients. 

Comparisons of sCD206 and sFAP-α levels in vitreous samples from control patients (n = 30), PDR patients (n = 38) and PVR patients (n = 10) were conducted with the Kruskal–Wallis test. The levels differed significantly between the three groups (*p* < 0.001 for both comparisons) (Table 2). Pairwise comparisons (Mann–Whitney test) indicated that sCD206 levels were significantly higher in patients with PDR and PVR than in control patients (*p* < 0.001; *p* = 0.001, respectively). Similarly, sFAP-α levels were significantly higher in patients with PDR and PVR than in control patients (*p* < 0.001; *p* = 0.033, respectively) (Table 2). A significant positive correlation (Spearman’s correlation coefficient) was found between vitreous fluid levels of sCD206 and levels of sFAP- α (r = 0.774; *p* < 0.001) (Figure 13).

### 2.10. Proinflammatory TNF-α and the Hypoxia Mimetic Agent CoCl_2_, but Not Proangiogenic VEGF Induce Upregulation of sFAP-α in Retinal Müller Glial Cells

From the immunohistochemical data presented above, the presence of sFAP-α in vitreous fluid may be derived from myofibroblasts and M2 macrophages transitioned to myofibroblasts. However, additional cellular sources could contribute to the measured sFAP-α. We investigated the effect of various proliferative vitreoretinal disorder-associated mechanisms, such as ischemic hypoxia, proinflammatory cytokines and proangiogenic factors, on the release of sFAP-α by HRMECs and by retinal Müller glial cells. With ELISA analysis of the culture medium, we demonstrated high constitutive sFAP-α production by Müller cells and weak production by HRMECs. The levels of sFAP-α were twelve-fold higher in Müller cells than from HRMECs. We also demonstrated that treatment of Müller cells with TNF-α and the hypoxia mimetic agent CoCl_2_ induced significant upregulation of sFAP-α in the culture medium compared to untreated control. However, the proangiogenic factor VEGF did not affect sFAP-α production compared to untreated control. Co-treatment with TNF-α plus the NF-ĸß inhibitor BAY11-7085 significantly attenuated TNF-α-induced upregulation of sFAP-α in Müller cells. However, BAY11-7085 did not affect CoCl_2_-induced upregulation of sFAP-α (Figure 14). In HRMECs, treatment with CoCl_2_, TNF-α or VEGF did not affect sFAP-α expression compared to untreated control.

## 3. Discussion

The key findings of our study were that in epiretinal membranes from patients with proliferative vitreoretinal disorders, most of the CD68^+^ monocytes/macrophages exhibit an M2 phenotype. In addition, we demonstrated that epiretinal membranes from patients with PVR were characterized by the presence of heavy pigmentation. The heavily pigmented cells were negative for the RPE marker pan-cytokeratin, but were positive for CD68 and CD206. These findings indicate that the heavily pigmented cells are monocytes/macrophages engulfing pigments rather than retinal RPE cells [13,14]. 

Most importantly, we demonstrated the existence of MMT in epiretinal membranes from patients with PDR and PVR. We identified cells undergoing macrophage-to-myofibroblast transition by the co-expression of the macrophage marker CD68 and the fibroblast/myofibroblast markers α-SMA and FSP-1. CD68^+^/FAP-α^+^ cells were not detected, and our data are in agreement with a study demonstrating FAP-α expression by smooth muscle cells, but not by CD68^+^ macrophages in human atherosclerotic plaques [25]. Further analysis revealed that the CD206^+^ M2 macrophages co-expressed α-SMA, FSP-1 and FAP-α, suggesting that MMT cells had a predominant CD206^+^ M2 phenotype (Figure 15). In studies of other fibrotic disorders, the CD206^+^ subset of M2 macrophages is strongly associated with fibrosis, and the MMT cells are largely derived from the CD206^+^ subset of M2 macrophages [6,7,8,17,18,20,21,26,27,28,29]. This may demonstrate that MMT cells transitioned through an M2 state before becoming myofibroblasts. We also identified CD206^+^ M2 macrophages with a spindle-like morphology, suggesting that they were in a transitional state. These findings were supported by double-labeling experiments identifying CD206^+^/FAP-α^+^ spindle-shaped cells. Our findings were recently supported by Laich and colleagues [30], who demonstrated the presence of CD206^+^ M2 monocytes/macrophages and Iba1^+^ myeloid cells in PVR epiretinal membranes. In addition, the majority of α-SMA^+^ cells co-expressed Iba-1, the leukocytes common antigen CD45 and the M2 monocyte/macrophage marker CD163. Furthermore, cultured vitreal hyalocytes from patients with PDR co-expressed the myeloid cell marker Iba-1 and α-SMA, suggesting that hyalocytes are capable of transdifferentiation to myofibroblasts during the course of PDR [15]. We cannot make definite conclusions on the specific myeloid origin of the CD206^+^ cells, since the CD68 marker used cannot discriminate between different myeloid cells (retinal microglia, infiltrating monocytes, in situ proliferating macrophages, perivascular macrophages). In mice where PVR was induced by intravitreal injection of RPE cells, CD206^+^ M2-type macrophages were also found in preretinal fibrous membranes [31].

The monocytes that transit into myofibroblasts during the MMT process need to be recruited to the ocular microenvironment. Specifically, in patients with proliferative vitreoretinal disorders, monocyte recruitment is induced by upregulation of the expression of monocyte chemotactic protein-1 (CCL2/MCP-1) [32], and activation may occur through elevated expression of scavenger receptor for phosphatidyl serine and oxidized low-density lipoprotein (CXCL16/SR-PSOX) [33]. These chemokine–receptor interactions (with CCR2 and CXCR6, respectively) mediate persistent mononuclear infiltration that leads to chronic inflammation, neovascularization and fibrosis [32,33].

The roles of TGF-ß_1_/ß-catenin in mediating pro-fibrotic signaling pathways [6,7,8,16,17] and in the MMT process are established [21,22,23,24]. In vitro studies demonstrated that TGF-ß_1_ treatment induced the transition of cultured macrophages into collagen-producing myofibroblasts, as shown by the expression of α-SMA (CD68^+^ α-SMA^+^ cells) [16,21]. In addition, most bone marrow macrophages that transition into α-SMA^+^ myofibroblasts expressed the M2 marker CD206 [21]. We demonstrated clinically extensive expression of TGF-ß_1_ and ß-catenin in vascular endothelial cells and myofibroblasts and the colocalization of ß-catenin and CD206^+^ in M2 macrophages. In analogy, in another fibrotic disease, ß-catenin is expressed in myofibroblasts in lung samples from patients with idiopathic pulmonary fibrosis [24]. Increased expression of ß-catenin protein in CD206^+^ M2 macrophages was preclinically observed in a model of lung cancer with similar polarization of M2 macrophages [34]. Our findings provide evidence of the involvement of the TGF-ß_1_/ß-catenin signaling pathway in fibrosis associated with proliferative vitreoretinal disorders. 

In addition to fibrosis enhancement, it is known that CD206^+^ M2 macrophages secrete proangiogenic factors and matrix metalloproteinases, promoting tumor angiogenesis and progression [34,35,36,37], thereby suggesting that the inhibition of M2 macrophage polarization may inhibit these processes [34,35,37]. Indeed, in vitro and in vivo studies demonstrated that suppressing the polarization of M2 macrophages was effective in inhibiting fibrosis in animal models of liver [27] and pulmonary [28] fibrosis. We demonstrated previously significant correlations between the numbers of CD68^+^ monocytes/macrophages [1] and CD163^+^ cells [2] and the degree of angiogenic activity in PDR epiretinal fibrovascular membranes. 

FAP-α is a type II transmembrane serine protease and a member of the prolyl peptidase family. Tissue FAP-α expression is normally low or undetectable but is selectively induced at sites of active tissue remodeling and repair. FAP-α is actively induced in myofibroblasts in fibrotic conditions and marks myofibroblast activation. In cancer, FAP-α has been reported to induce tumor angiogenesis, growth and invasion. Membrane-bound FAP-α may be shed as an active soluble form (sFAP-α), which is referred to as α2-antiplasmin cleaving enzyme [11,12]. Similarly, CD206 is shed from activated macrophages by proteases generating a soluble form of CD206 (sCD206). sCD206 is a macrophage activation marker and a potential biomarker for inflammation, fibrosis, cancer and infection [38,39]. We demonstrated that the levels of sCD206 and sFAP-α were significantly upregulated in vitreous fluid samples from patients with proliferative vitreoretinal disorders and that there was a significant positive correlation between their levels. From immunohistopathological data, sFAP-α in the vitreous may be derived from M2 macrophages transitioned to myofibroblasts and myofibroblasts. We also showed that cultured human Müller cells produce sFAP-α. These results provided additional evidence that macrophage and myofibroblast activation may contribute to the pathogenesis of proliferative vitreoretinal disorders and that inflammation and fibrosis are interrelated events in these disorders. In addition, our findings suggest that sCD206 and sFAP-α are potential biomarkers for proliferative vitreoretinal disorders. 

To the best of our knowledge, our study is the first to demonstrate the expression of CD206 by retinal endothelial cells. Moreover, our observations are in line with a previous report [40]. Indeed, dermal microvascular endothelial cells displayed 50–95% CD206 expression, which diminished with increasing passage number and was unrecoverable following cytokine treatment [40]. It is likely that the expression of CD206, observed within the vascular endothelial cells lining the new blood vessels in epiretinal membranes from patients with PDR, is present during a transitory state and will diminish over time. On the contrary, a particular in vivo milieu might be required for endothelial cells to maintain their expression of CD206. 

Inflammation is a key feature of proliferative vitreoretinal disorders. HRMECs and retinal Müller glial cells are major cell types in inflammatory reactions in the retina [1,3,5,33]. We demonstrated high constitutive sFAP-α production by Müller cells and much lower levels in HRMEC cultures. We also demonstrated the capability of TNF-α and hypoxia (but not VEGF) to induce the production of sFAP-α in Müller cells. Our findings are in line with a previous study documenting the capacity of TNF-α to upregulate sFAP-α in cultures of human aortic smooth muscle cells [25]. Inhibition of the inflammatory transcription factor NF-ĸß significantly attenuated TNF-α-induced upregulation of sFAP-α in the culture medium of Müller cells.

Our immunohistochemical analysis provided evidence for the occurrence of the MMT process in the intraocular microenvironment of patients with proliferative vitreoretinal disorders. However, future investigations and alternative techniques are needed to further explore the role of the MMT process in promoting myofibroblast formation in these disorders. For instance, cell lineage tracing studies by adoptive transfer of dye-labeled monocytes in experimental models [21] or single-cell RNA sequencing and fate-mapping studies on human epiretinal membranes could provide additional information on the involvement of MMT in proliferative vitreoretinal disorders. 

## 4. Materials and Methods

### 4.1. Vitreous Humor Samples and Epiretinal Membrane Specimens

Undiluted vitreous fluid samples (200–300 µL) were obtained from 38 patients with PDR during pars plana vitrectomy for the treatment of tractional retinal detachment and/or nonclearing vitreous hemorrhage. Vitreous fluid samples were also obtained from 10 patients with PVR during vitreoretinal surgery for the treatment of primary rhegmatogenous retinal detachment complicated by PVR. A set of control samples was obtained from 30 patients who had undergone vitrectomy for the treatment of rhegmatogenous retinal detachment without PVR. Control subjects were clinically checked to be free from diabetes or other systemic disease. All these samples were processed as described previously [1,2]. 

Epiretinal fibrovascular membranes were obtained from 12 patients with PDR during pars plana vitrectomy for the repair of tractional retinal detachment. Epiretinal fibrocellular membranes were also obtained from 12 patients without diabetes undergoing vitreoretinal surgery for the treatment of retinal detachment complicated by PVR. These epiretinal membrane samples were processed as described previously [1,2,3,4]. The membranes were fixed for 2 h in a 10% formalin solution and embedded in paraffin; 3 µm sections were cut off these paraffin blocks using a Microm HM 360 Microtome (Thermo Scientific, Waltham, MA, USA). The sections were dried overnight at 52 °C. Before staining, the following dewaxing protocol was used: 3 consecutive 2 min incubations in xylene, followed by 3 consecutive 2 min treatment with 100% ethanol.

### 4.2. Immunohistochemical Staining of Human Epiretinal Membranes

For CD31 and α-SMA detection, antigen retrieval was performed by boiling the sections in citrate-based buffer (pH 5.9–6.1) (BOND Epitope Retrieval Solution 1; Leica) for 10 min. For CD68, CD206, CD86, iNOS, FAP-α, FSP-1, TGF-ß1, ß-catenin and cytokeratin detection, antigen retrievals were performed by boiling the sections in Tris/EDTA buffer (pH 9) (BOND Epitope Retrieval Solution 2; Leica) for 20 min. Subsequently, the sections were incubated for 60 min as described previously [1,2,3,4] with the antibodies listed in Table 3. 

To identify the phenotype of cells expressing CD68 and CD206, sequential double immunohistochemistry was performed. First, primary antibodies (anti-CD68 or anti-CD206) were applied and subsequently reacted with peroxidase-conjugated secondary antibody to define monocytes/macrophages by enzymatic reaction of the 3,3′-diaminobenzidine tetrahydrochloride substrate, yielding brown precipitates. Thereafter, incubation with either anti-α-SMA, anti-FAP-α, anti-FSP-1 or anti-ß-catenin was followed by treatment with alkaline phosphatase-conjugated secondary antibody and Fast Red reactions. No counterstain was applied. In negative controls, the incubation step with primary antibodies was omitted from the staining protocol. Instead, only the ready-to-use DAKO Real antibody Diluent (Agilent Technologies Product Code 52022) was applied.

CD68^+^ and CD206^+^ cells were counted in 5 representative fields with the use of an eyepiece with a calibrated grid, in combination with a 40× objective. These representative fields were selected based on the presence of immunoreactive cells. With the used magnification and calibration, an area of 0.33 mm × 0.2 mm was evaluated. The percentages of the numbers of CD68^+^ cells and CD206^+^ cells co-expressing the myofibroblast markers α-SMA, FSP-1 and FAP-α to the total number of CD68^+^ cells and CD206^+^ cells, respectively, were calculated. We report both the global percentage based on the counting for all samples combined and the range (percentage calculations of individual samples).

### 4.3. Human Retinal Müller Glial Cell and Human Retinal Microvascular Endothelial Cell Cultures 

Human retinal Müller glial cells (MIO-M1) (a generous gift from Prof. A. Limb, Institute of Ophthalmology, University College, London, UK) and human retinal microvascular endothelial cells (HRMECs) (Cell Systems Corporation, Kirkland, WA, USA) were cultured as described previously [1]. 

For HRMEC stimulation experiments, 6-well plates (TPP, Trasadingen, Switzerland) were coated with 0.1% (*w*/*v*) gelatin for a period of 1 h at room temperature. Afterward, excess gelatin coating was removed, and HRMECs were seeded at a density of 140,000 cells/well in EBM-2 medium supplemented with EGM 2-MV BulletKit (Lonza, Basel Switzerland). Upon adherence, HRMECs were treated with EBM-2 medium containing 3% (*v*/*v*) fetal calf serum (FCS) supplemented with VEGF (30 ng/mL) to maintain the endothelial phenotype. The medium was refreshed after two days. On day four, the cells were processed for flow cytometry staining.

The following stimuli were used to investigate regulation of expression of sFAP-α in HRMECs and Müller cells: 50 ng/mL recombinant human tumor necrosis factor-alpha (TNF-α) (Cat No 210-TA, R&D Systems, Minneapolis, MN, USA), 50 ng/mL recombinant human vascular endothelial growth factor (VEGF) (Cat No 293-VE-050, R&D Systems) or 300 µM of the hypoxia mimetic agent cobalt chloride (CoCl_2_) (AVONCHEM Limited, Nacclesfield, Cheshire, UK) in the absence or presence of the nuclear factor-kappa B (NF-κB) inhibitor BAY11-7085 (10 µM) (Cat No sc-202490, Santa Cruz Biotechnology Inc., Santa Cruz, CA, USA). After 24 h, cell supernatants were collected and processed for ELISA analysis. 

### 4.4. Enzyme-Linked Immunosorbent Assay

Enzyme-linked immunosorbent assay (ELISA) kits for human soluble FAP-α (sFAP-α) (Cat No DY3715) and human soluble CD206 (sCD206) (Cat No ab277420) were purchased from R&D Systems and Abcam, respectively. Levels of human sFAP-α and sCD206 in vitreous fluid and in cell culture medium were determined according to the manufacturer’s instructions. The minimum detection limits for sFAP-α and sCD206 ELISA kits were 25 pg/mL and 0.41 ng/mL, respectively.

### 4.5. Flow Cytometry

HRMECs were detached using the ReagentPack subculture kit from Lonza. Firstly, the medium was removed, and the cells were washed with HEPES pre-warmed to 37 °C. Thereafter, trypsin, pre-warmed to 37 °C, was added to the cells for a period of 1 min, detaching the cells and finally, trypsin-neutralizing solution was added. The cells were allowed to recover for a period of 1 h at room temperature in medium supplemented with 10% (*v*/*v*) FCS. To exclude dead cells from the analysis, cells were incubated with Zombie Aqua viability dye (BioLegend, San Diego, CA, USA) in PBS for 15 min at room temperature. In addition, the cells were simultaneously treated with human FcR blocking reagent (130-059-901; Miltenyi Biotech, Westphalia, Germany). Afterward, the cells were stained with the Fluorescein isothiocyanate-labeled mouse anti-human CD206 (catalog number 551135 clone 19.2; BD Biosciences, San Jose, CA, USA) in fluorescence-activated cell sorting (FACS) buffer (PBS, 2% (*v*/*v*) FCS, 2 mM ethylenediamine tetraacetic acid) for 30 min on ice. The cells were subsequently washed twice in FACS buffer and fixed with 0.4% (*v*/*v*) formaldehyde in FACS buffer. Data acquisition was carried out using an LSRFortessa X-20 cell analyzer (BD Biosciences), and data analysis was conducted with FlowJo software, version 10.8.1. (Tree Star, Ashland, OR, USA). 

### 4.6. Statistical Analysis

Data were collected, stored and managed in a spreadsheet using Microsoft Excel 2010^®^ software. Data were analyzed and figures prepared using SPSS^®^ version 21.0 (IBM Inc., Chicago, IL, USA) or GraphPad software, v9.5.1 (GraphPad Software Inc. La Jolla, CA, USA). Tests for normality were conducted using Shapiro–Wilk test and Q-Q plots. Data were not normally distributed and presented using box and whisker plots showing the medians, upper and lower quartiles and range. Consequently, Kruskal–Wallis and Mann–Whitney tests (applying Bonferroni correction where necessary) were used to test the differences between the groups. Additionally, Spearman’s correlation analysis was carried out. Any output with a *p*-value below 0.05 was interpreted as an indicator of statistical significance.

## 5. Conclusions

Our findings indicate that the MMT process contributes to myofibroblast formation in epiretinal membranes, contributing to the development and progression of fibrosis in proliferative vitreoretinal disorders. This transition involves macrophages with a predominant M2 phenotype. Blocking the recruitment of mononuclear-derived macrophages, inhibiting the polarization of M2 macrophages or targeting macrophage signaling pathways may be beneficial for the prevention of fibrosis in proliferative vitreoretinal disorders. This should be the subject of future studies.

## Figures and Tables

**Figure 1 ijms-24-13510-f001:**
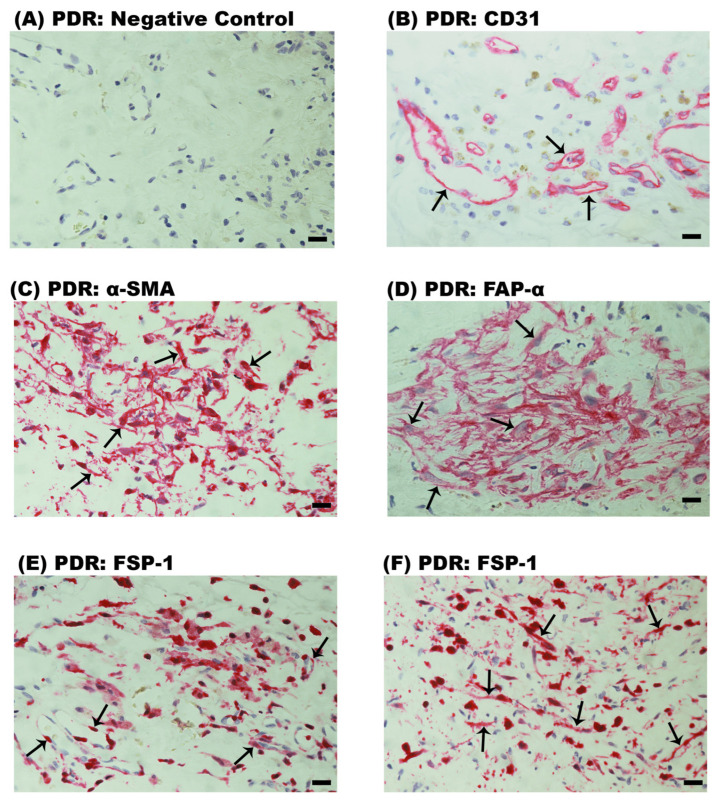
Immunodetection of endothelial and mesenchymal markers in proliferative diabetic retinopathy epiretinal fibrovascular membranes. The figure shows representative immunohistochemical analysis results for 1 patient out of a total of 12 investigated patients. Each panel is from a different patient (**A**). Negative control slide showing no labeling. (**B**) Immunohistochemical staining for the endothelial cell marker CD31 demonstrating pathologic new blood vessels (arrows). Immunohistochemical staining for (**C**) α-smooth muscle actin (α-SMA) and (**D**) fibroblast activation protein-α (FAP-α) showing immunoreactivity in stromal spindle-shaped myofibroblasts (arrows). Immunohistochemical staining for fibroblast specific protein-1 (FSP-1) showing immunoreactivity in (**E**) vascular endothelial cells lining new blood vessels (arrows) and (**F**) stromal spindle-shaped cells (arrows) (scale bar, 10 µm).

**Figure 2 ijms-24-13510-f002:**
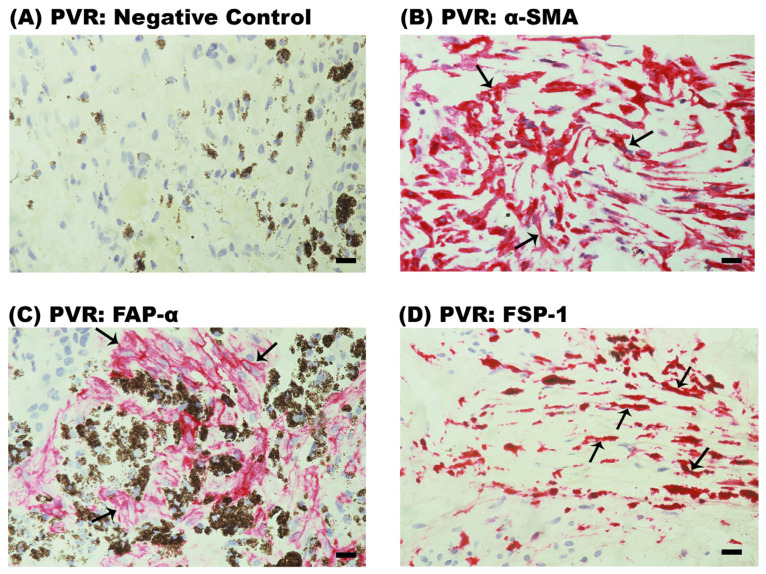
Immunodetection of mesenchymal markers in proliferative vitreoretinopathy epiretinal fibrocellular membranes. Representative figures are provided for 1 patient out of a total of 12 patients, and each panel is from a different patient. Note that PVR membranes are characterized by a high degree of pigmentation. (**A**) Negative control slide showing no labeling. (**B**) Immunohistochemical staining for α-smooth muscle actin (α-SMA) showing immunoreactivity in myofibroblasts (arrows). Immunohistochemical stainings for (**C**) fibroblast activation protein-α (FAP-α) (arrows) and (**D**) fibroblast specific protein-α (FSP-α) (arrows) showing immunoreactivity in spindle-shaped myofibroblast-like cells (scale bar, 10 µm).

**Figure 3 ijms-24-13510-f003:**
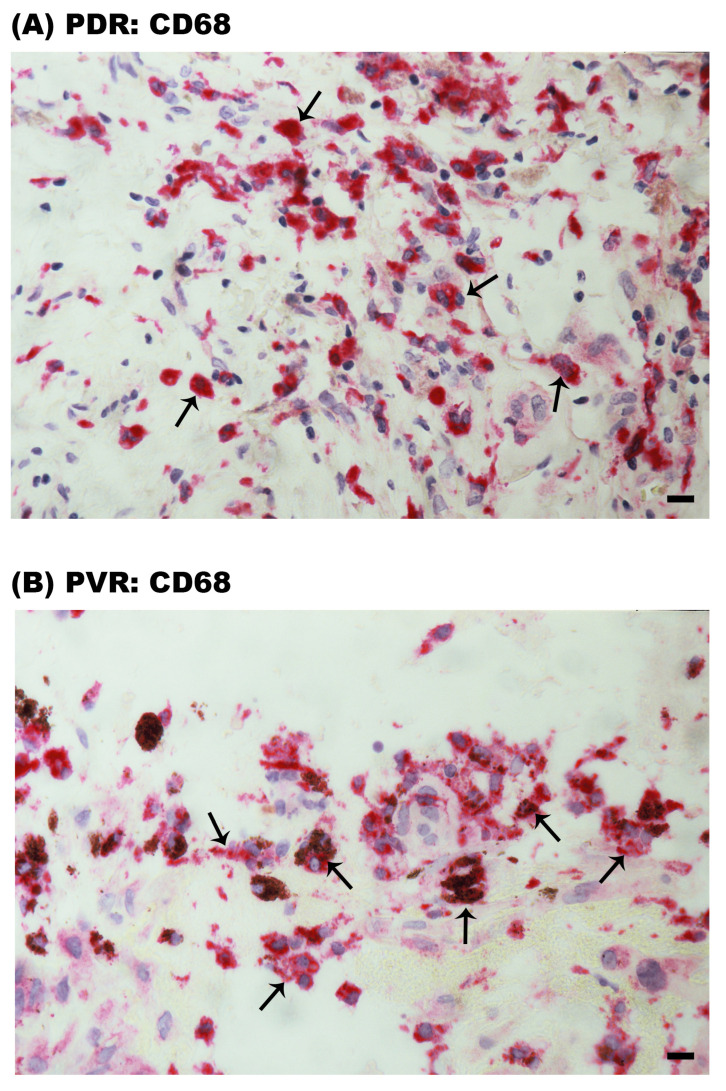
Identification of CD68^+^ cells in epiretinal membranes. Immunohistochemical staining for the monocyte/macrophage marker CD68 showing infiltrating monocytes/macrophages in (**A**) a membrane from a patient with proliferative diabetic retinopathy (PDR) (arrows) and in (**B**) a membrane from a patient with proliferative vitreoretinopathy (PVR) (arrows). Note that some of the CD68^+^ cells in the PVR membrane contain pigment. Representative figures are provided for 1 patient out of a total of 12 PDR or 12 PVR patients (scale bar, 10 µm).

**Figure 4 ijms-24-13510-f004:**
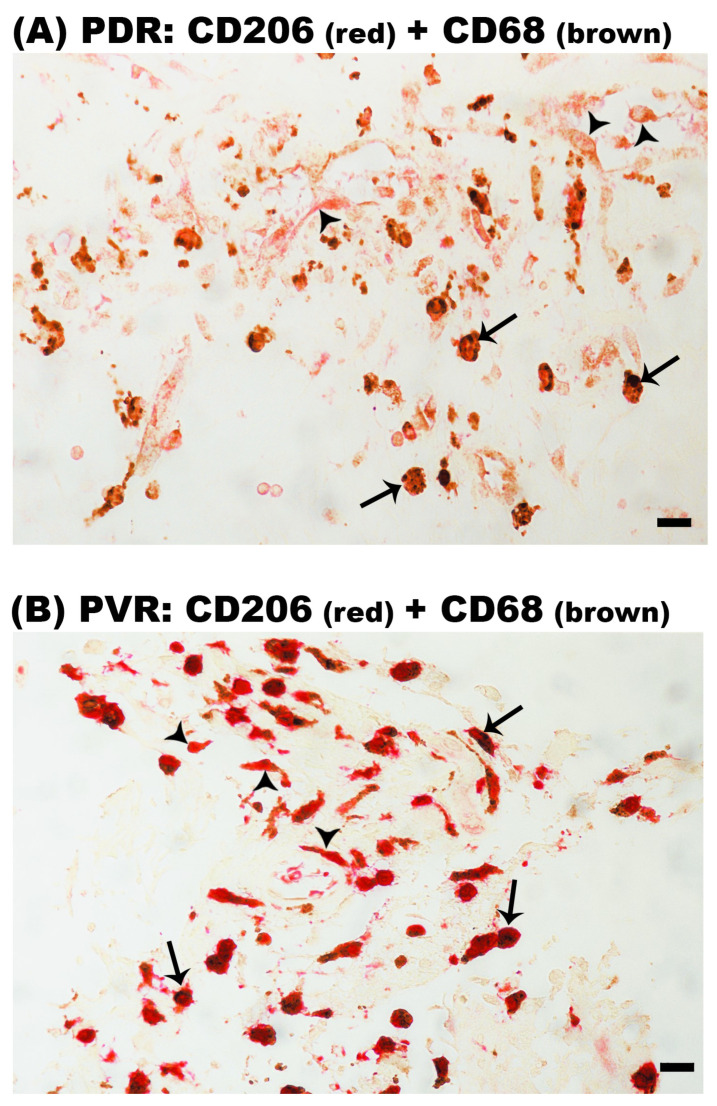
Identification of CD68^+^/CD206^+^ double-positive cells in epiretinal membranes. Double immunohistochemical staining for CD68 (brown) and CD206 (red) demonstrating co-expression in monocytes/macrophages in a membrane from a patient with proliferative diabetic retinopathy (PDR) (arrows) (**A**) and in a membrane from a patient with proliferative vitreoretinopathy (PVR) (arrows) (**B**). Note the presence of CD206^+^ single-positive cells (arrowheads). No counterstain to visualize the cell nuclei was applied (scale bar, 10 µm).

**Figure 5 ijms-24-13510-f005:**
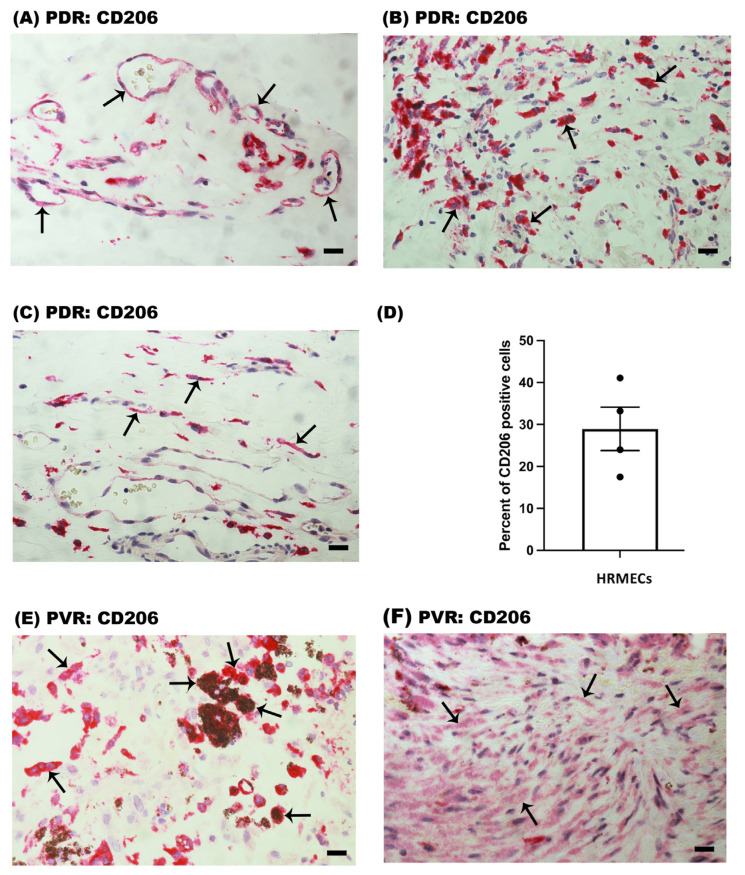
Identification of CD206^+^ cells in epiretinal membranes. Immunohistochemical staining for the M2 macrophage marker CD206 showing immunoreactivity in (**A**) vascular endothelial cells (arrows), (**B**) stromal monocytes/macrophages (arrows) and (**C**) stromal spindle-shaped cells (arrows) in membranes from patients with proliferative diabetic retinopathy (PDR). (**D**) CD206 surface expression was detected in human retinal microvascular endothelial cells (HRMECs) using flow cytometry. Results are presented as the mean percent of CD206^+^ cells ± SEM and are derived from four independent experiments. Immunoreactivity for CD206 was detected (**E**) in monocytes/macrophages (arrows) and (**F**) spindle-shaped cells (arrows) in membranes from patients with proliferative vitreoretinopathy (PVR). Note that some of the CD206^+^ cells in the PVR membrane contained pigment. Representative figures are provided for 1 patient out of a total of 12 PDR or 12 PVR patients. Each panel is from a different patient (scale bar, 10 µm).

**Figure 6 ijms-24-13510-f006:**
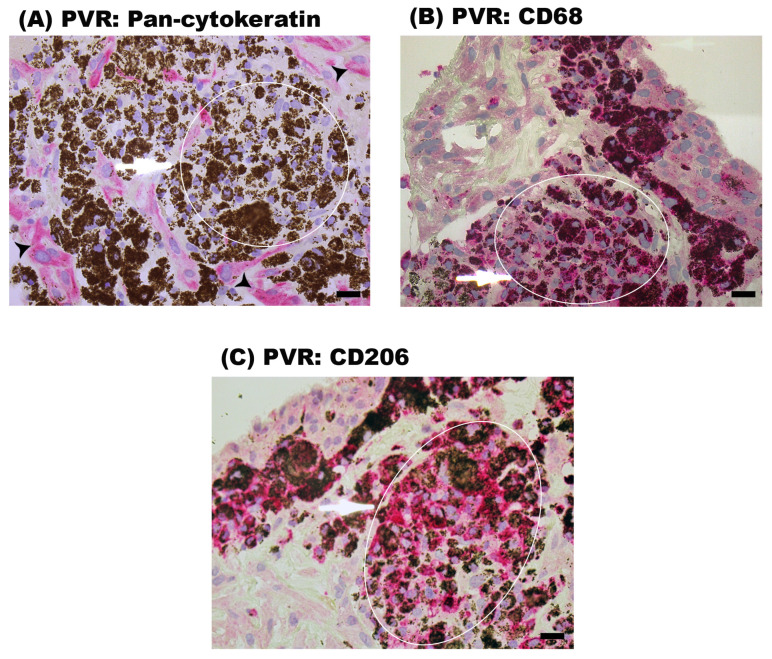
Immunohistochemical staining of proliferative vitreoretinopathy (PVR) epiretinal fibrocellular membranes. Immunohistochemical staining for (**A**) pan-cytokeratin, (**B**) CD68 and (**C**) CD206 of serial sections. The white circle and ellipses outline the same area of interest present in the different serial sections. PVR membranes contained cells that were immunoreactive for pan-cytokeratin (arrowheads) (**A**). Note that the heavily pigmented cells were not immunoreactive for pan-cytokeratin (arrow) (**A**). The heavily pigmented cells expressed CD68 (arrow) (**B**) and CD206 (arrow) (**C**) (scale bar, 10 µm). Representative figures are provided for 1 patient out of a total of 12 PVR patients.

**Figure 7 ijms-24-13510-f007:**
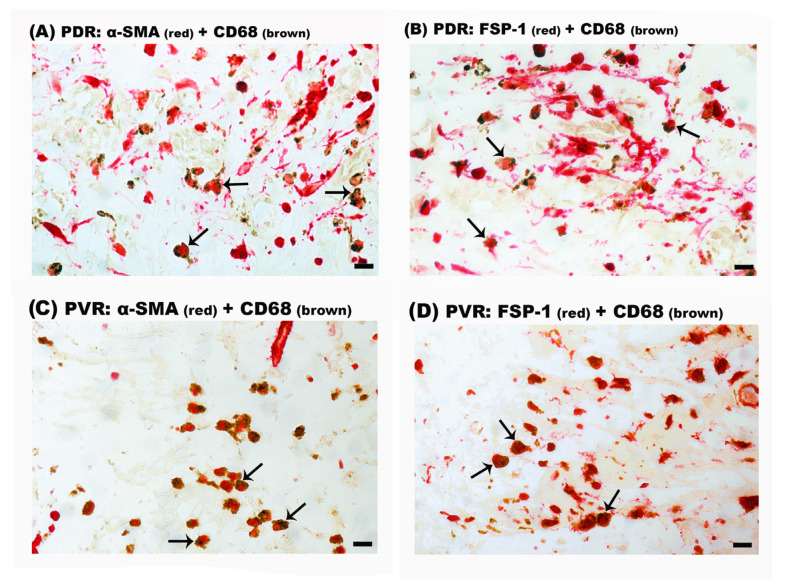
Identification of macrophage–myofibroblast transition cells in epiretinal membranes. Double immunohistochemical analysis for co-expression of the monocyte/macrophage marker CD68 (brown) and either α-smooth muscle actin (α-SMA) (red) or fibroblast-specific protein-1 (FSP-1) (red). CD68^+^/α-SMA^+^ double positive cells were identified in (**A**) a membrane from a patient with proliferative diabetic retinopathy (PDR) (arrows) and in (**C**) a membrane from a patient with proliferative vitreoretinopathy (PVR) (arrows). CD68^+^/FSP-1^+^ double positive cells were identified in (**B**) a membrane from a patient with PDR (arrows) and in (**D**) a membrane from a patient with PVR (arrows). No counterstain to visualize the cell nuclei was applied (scale bar, 10 µm).

**Figure 8 ijms-24-13510-f008:**
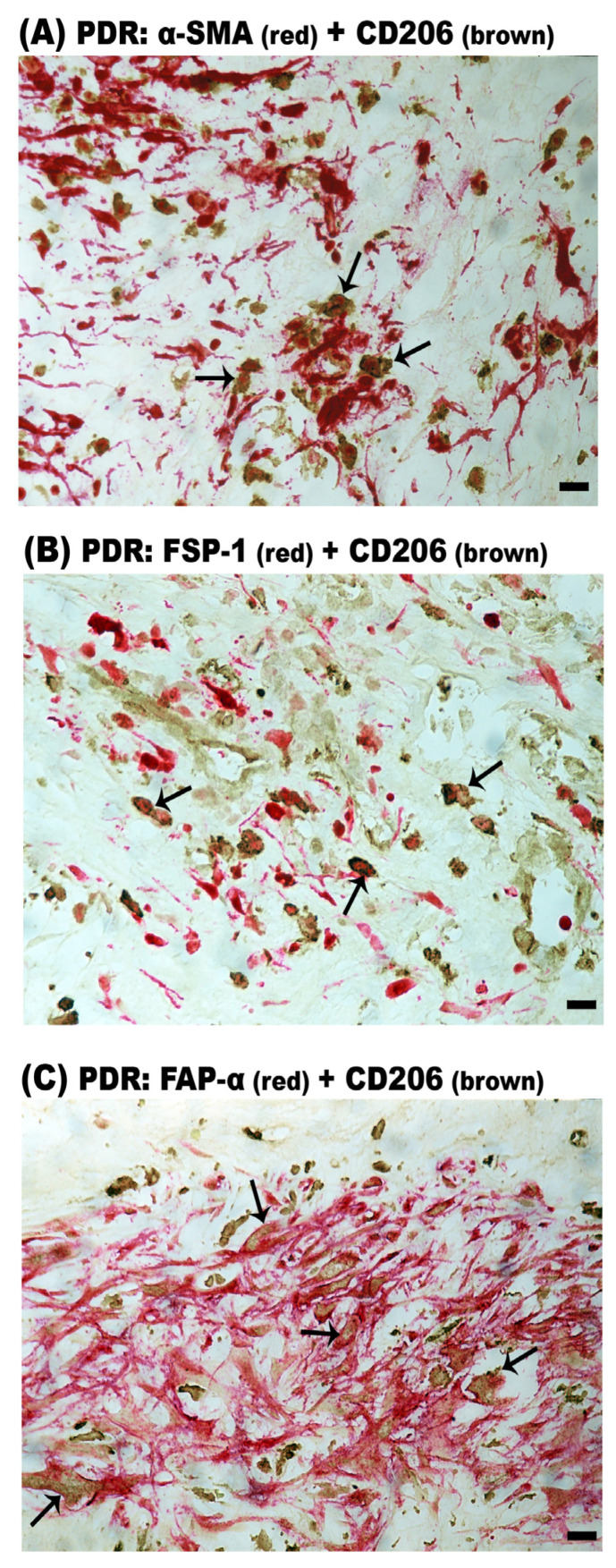
CD206^+^ cells co-express myofibroblast markers in epiretinal membranes from patients with proliferative diabetic retinopathy (PDR). Double immunohistochemistry for CD206 (brown) and either α-smooth muscle actin (α-SMA) (red), fibroblast specific protein-1 (FSP-1) (red) or fibroblast activation protein-α (FAP-α) (red). (**A**) CD206^+^/α-SMA^+^ cells (arrows), (**B**) CD206^+^/FSP-1^+^ cells (arrows) and (**C**) CD206^+^/FAP-α^+^ cells (arrows) were identified. No counterstain to visualize the cell nuclei was applied (scale bar, 10 µm).

**Figure 9 ijms-24-13510-f009:**
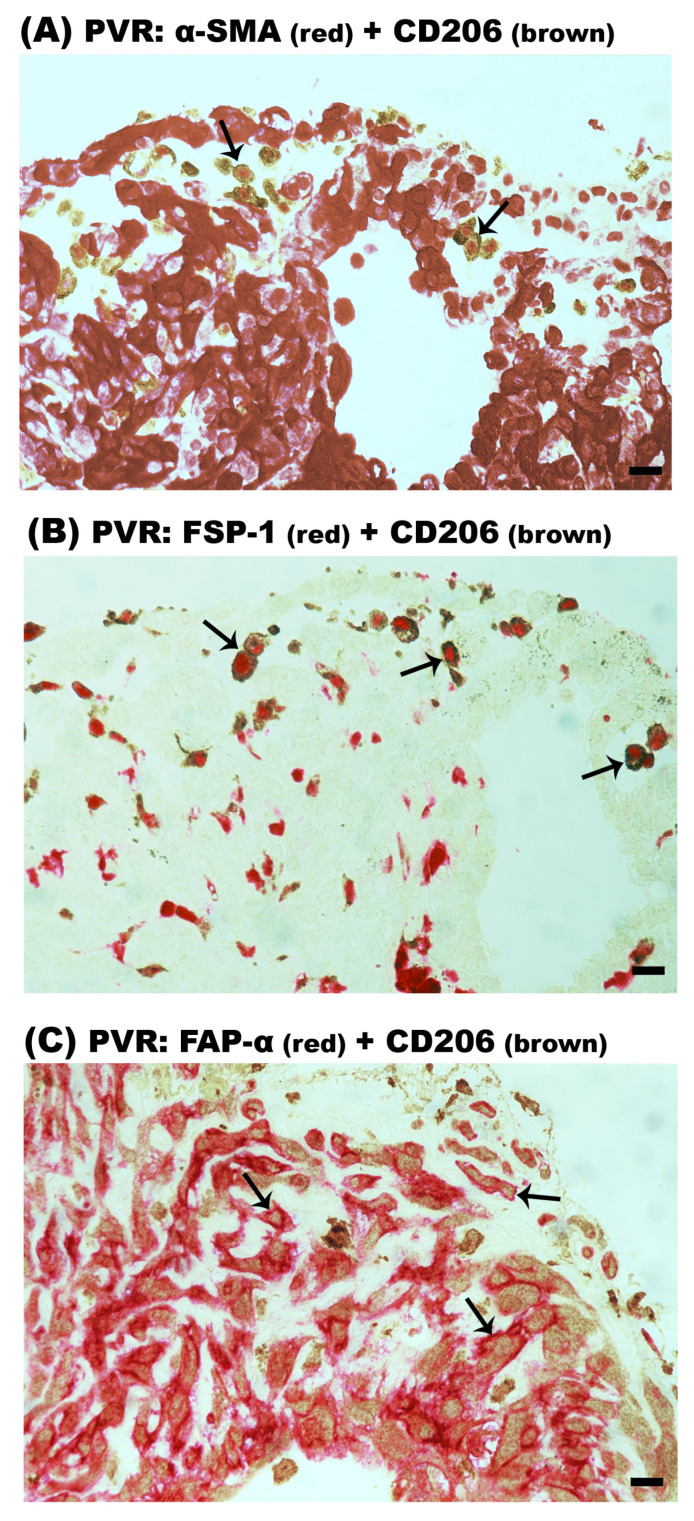
CD206^+^ cells co-express myofibroblast markers in epiretinal membranes from patients with proliferative vitreoretinopathy (PVR). Double immunohistochemistry for CD206 (brown) and either α-smooth muscle actin (α-SMA) (red), fibroblast-specific protein-1 (FSP-1) (red) or fibroblast activation protein-α (FAP-α) (red). (**A**) CD206^+^/α-SMA^+^ cells (arrows), (**B**) CD206^+^/ FSP-1^+^ cells (arrows) and (**C**) CD206^+^/FAP-α^+^ cells (arrows) were identified. No counterstain to visualize the cell nuclei was applied (scale bar, 10 µm).

**Figure 10 ijms-24-13510-f010:**
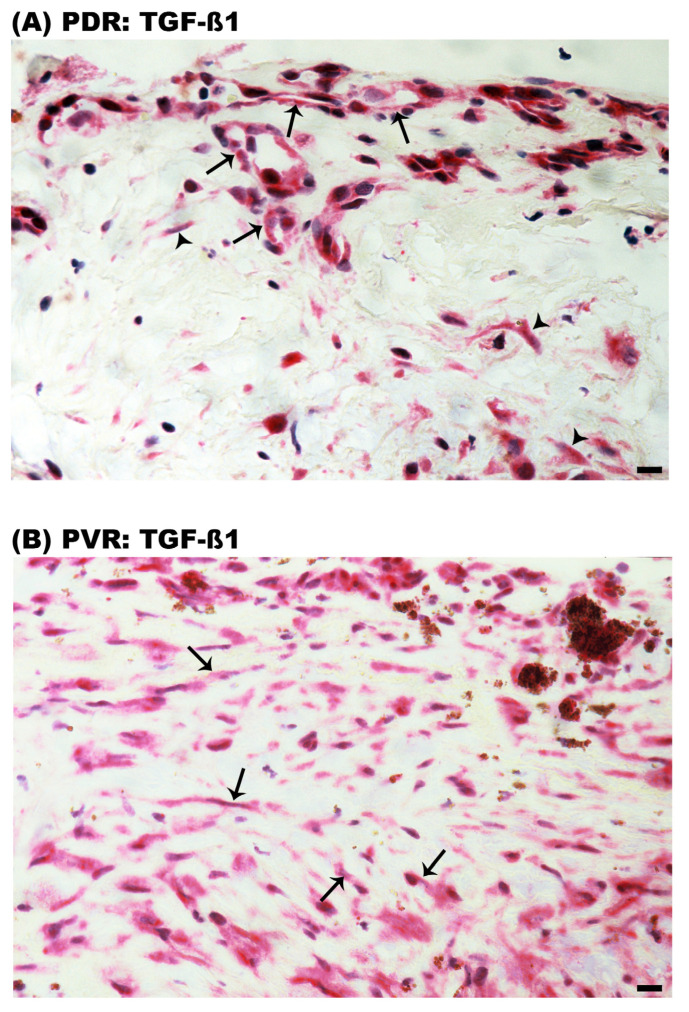
Expression of transforming growth factor-ß1 (TGF-ß1) in epiretinal membranes. Immunohistochemical staining for TGF-ß1 showed immunoreactivity in (**A**) vascular endothelial cells (arrows) and stromal spindle-shaped cells (arrowheads) in a membrane from a patient with proliferative diabetic retinopathy (PDR) and in (**B**) spindle shaped myofibroblast-like cells in a membrane from a patient with proliferative vitreoretinopathy (PVR) (arrows) (scale bar, 10 µm). Representative figures are provided for one patient out of a total of 12 PDR or 12 PVR patients.

**Figure 11 ijms-24-13510-f011:**
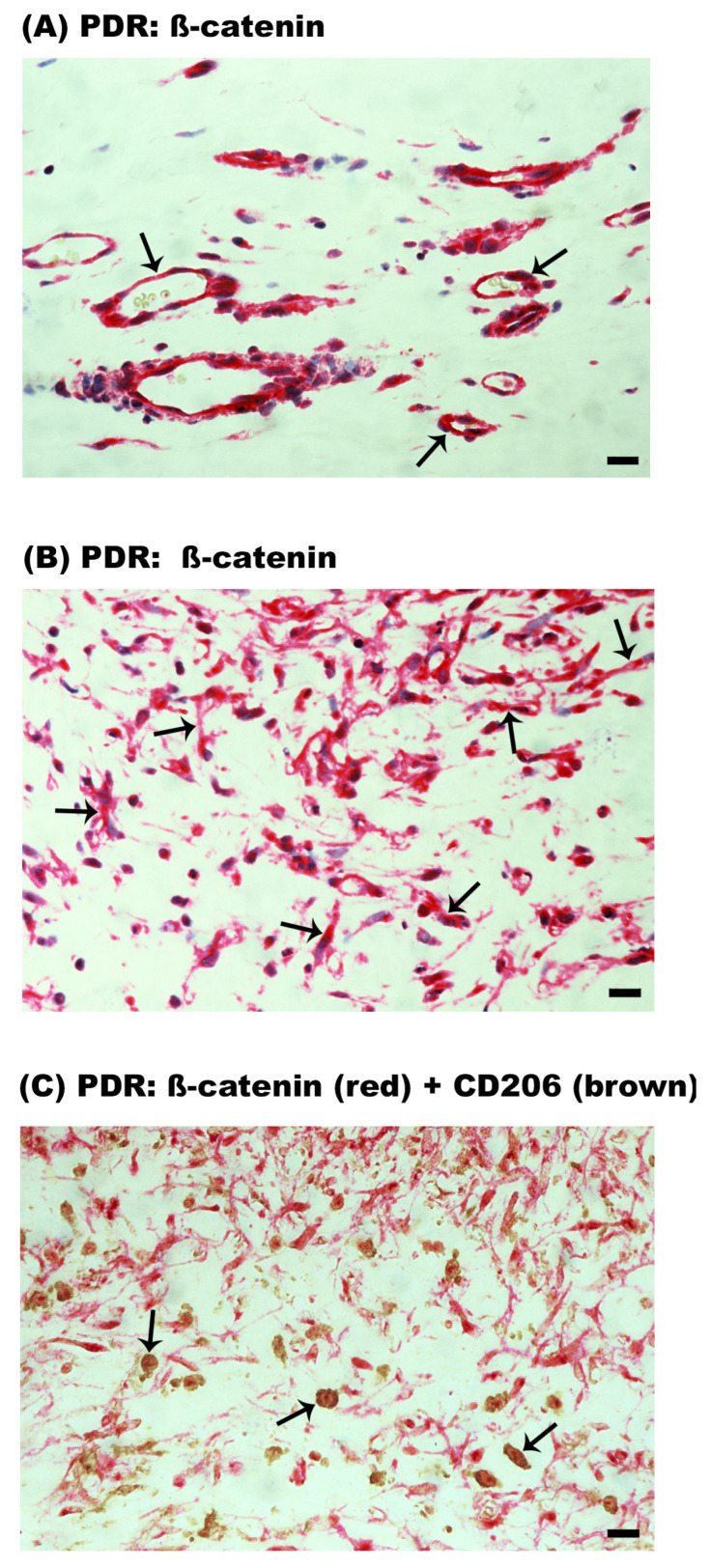
Expression of ß-catenin in epiretinal membranes from patients with proliferative diabetic retinopathy (PDR). Immunohistochemical staining for ß-catenin showed immunoreactivity in (**A**) vascular endothelial cells (arrows) and (**B**) stromal cells (arrows). (**C**) Double immunohistochemical staining for ß-catenin (red) and CD206 (brown) demonstrated co-expression in stromal cells (arrows). No counterstain to visualize the cell nuclei was applied (scale bar, 10 µm).

**Figure 12 ijms-24-13510-f012:**
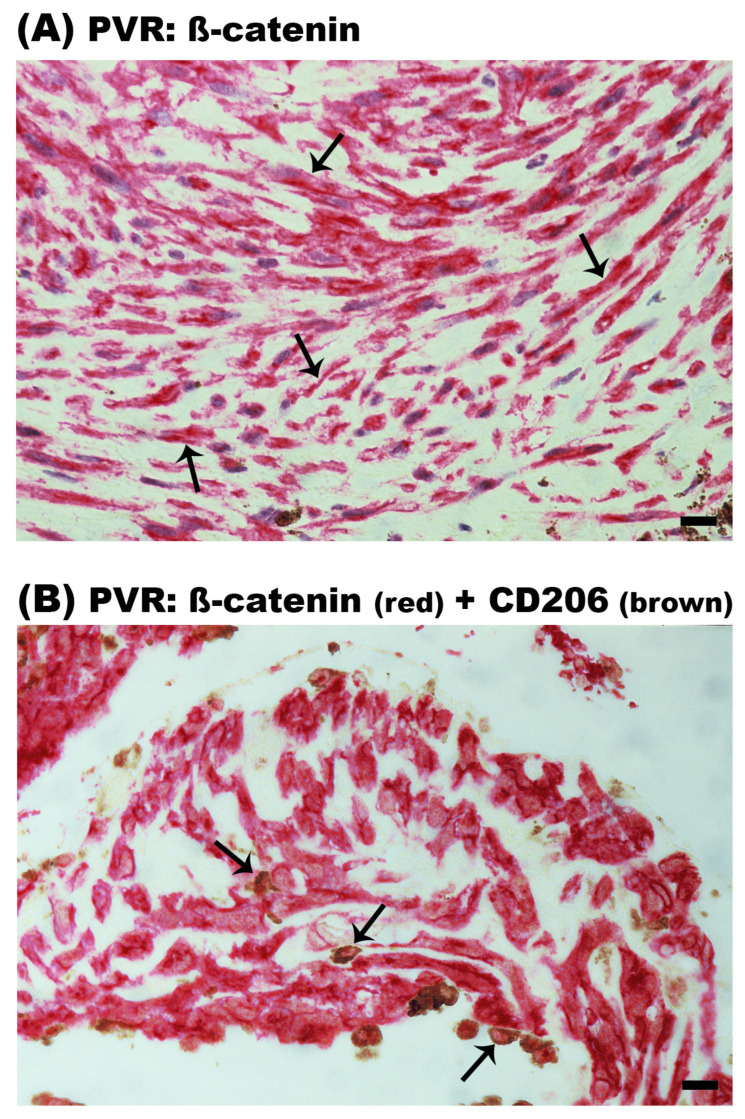
Expression of ß-catenin in epiretinal membranes from patients with proliferative vitreoretinopathy (PVR). Immunoreactivity in spindle-shaped myofibroblast-like cells is indicated by arrows (**A**), and ß-catenin and CD206 co-expression by arrows (**B**). No counterstain to visualize the cell nuclei was applied (scale bar, 10 µm).

**Figure 13 ijms-24-13510-f013:**
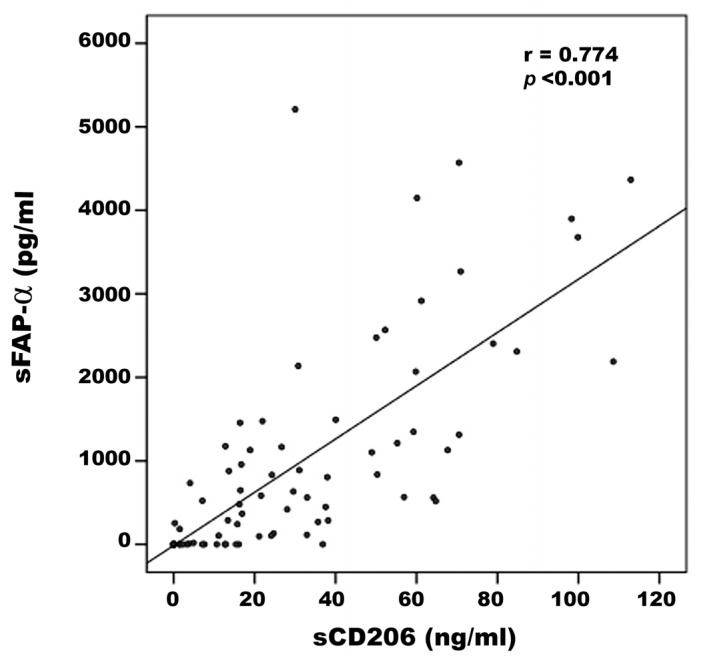
Significant positive correlation between vitreous fluid levels of soluble CD206 (sCD206) and soluble fibroblast activation protein-α (sFAP-α). A summary of the measured levels is shown in Table 2.

**Figure 14 ijms-24-13510-f014:**
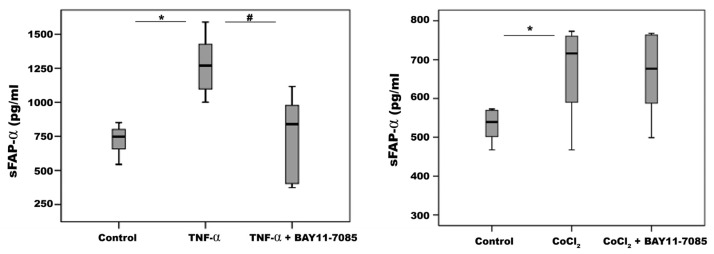
Human retinal Müller glial cells were left untreated or treated with tumor necrosis factor-α (TNF-α) (50 ng/mL), cobalt chloride (CoCl_2_) (300µM), TNF-α (50 ng/mL) plus BAY11-7085 (10 µM) or CoCl_2_ (300µM) plus BAY11-7085 (10 µM) for 24h. Levels of soluble fibroblast activation protein-α (sFAP-α) were quantified in the culture media by ELISA. Results are expressed as median (interquartile range) from three different experiments (each experiment: n = 6). Kruskal–Wallis test and Mann–Whitney tests were used for comparisons between three groups and two groups, respectively. * *p* < 0.05 compared with values obtained from untreated cells. # *p* < 0.05 compared with TNF-α plus BAY11-7085-treated cells.

**Figure 15 ijms-24-13510-f015:**
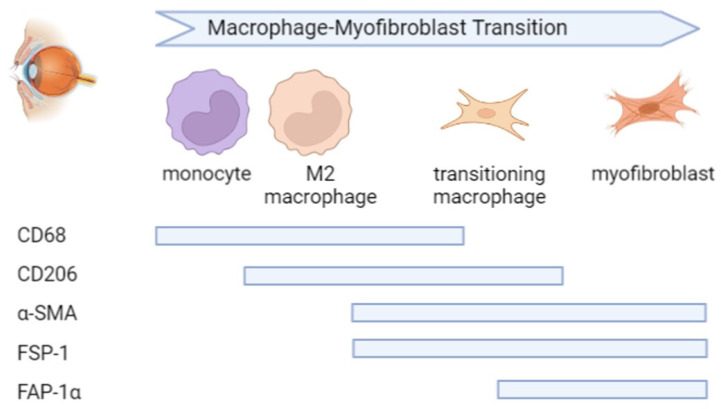
The process of macrophage–myofibroblast transition in proliferative vitreoretinal disorders. Based on our results, we propose the depicted sequence of changes in monocytes while they transition to myofibroblasts. First, macrophages acquire an M2 phenotype, and, at this stage, the expression of corresponding markers, such as CD206, is enhanced. Subsequently, myofibroblast-specific genes are turned on, and apparently, α-SMA and FSP-1 become detectable, whereas the expression of FAP-α remains precluded, as the latter marker was not costained with CD68. The figure is generated using BioRender.com.

**Table 1 ijms-24-13510-t001:** Summary of immunohistochemical findings.

Marker	PDR		PVR	
	Presence in Membranes *	Cell Type	Presence in Membranes	Cell Type
Pigment	-	NA	+	M/M
CD31	+	E	-	NA
CD68	+	M/M, MMT	+	M/M, MMT
CD206	+	M2, MMT, E	+	M2, MMT
CD86	-	NA	-	NA
iNOS	-	NA	-	NA
α-SMA	+	MMT, MF	+	MMT, MF
FAP-α	+	MMT. MF	+	MMT, MF
FSP-1	+	MMT. MF, E	+	MMT, MF
Pancytokeratin	-	NA	+	RPE
TGF-ß1	+	E, MF	+	MF
ß-catenin	+	E, MF, M2	+	MF, M2

* + present; - absent. E= endothelial cells of pathological neovessels; M2 = M2 macrophages, MF = myofibroblasts, M/M = monocytes/macrophages, MMT = cells co-expressing macrophage and myofibroblast markers; NA, not applicable; RPE = retinal pigment epithelial cells.

**Table 2 ijms-24-13510-t002:** Comparisons of soluble CD206 (sCD206) and soluble fibroblast activation protein-α (sFAP-α) in vitreous samples from control patients with rhegmatogenous retinal detachment (RD), proliferative diabetic retinopathy (PDR) patients and proliferative vitreoretinopathy (PVR) patients.

Disease Group	sCD206 (ng/mL)	sFAP-α (pg/mL)
RD (n = 30) Median (IQR)	12.63 (1.57–21.35)	57.00 (ND–308.25)
PDR (n = 38) Median (IQR)	32.01 (16.63–59.9)	1128.5 (549.5–2181.38)
PVR (n = 10) Median (IQR)	60.84 (12.35–100.88)	702.00 (ND–2777.00)
*p*-value (Kruskal–Wallis test)	<0.001 *	<0.001 *

* Statistically significant at 5% level of significance. IQR = interquartile range. ND = below detection limit

**Table 3 ijms-24-13510-t003:** Antibodies used in this study.

Primary Antibody	Dilution	Source *
Anti-CD31 (Clone JC70A) (mc)	ready-to-use	Dako
Anti-α-smooth muscle actin (clone 1A4) (mc)	ready-to-use	Dako
Anti-CD68 (Clone KP1) (mc)	ready-to-use	Dako
Anti-CD206 (Cat No MAB25341) (mc)	1/100	R & D Systems
Anti-CD86 (Cat No 91882) (mc)	1/100	Cell SignalingTechnology
Anti-iNOS (Cat No ab115819) (mc)	1/50	Abcam
Anti-FAP- α (Cat No ab207178) (mc)	1/250	Abcam
Anti-FSP-1 (Cat No ab124805) (mc)	1/500	Abcam
Anti-TGF- ß1 (Cat No SAB4502954)	1/50	Sigma-Aldrich
Anti-ß-catenin (mc)	Ready-to-use	Agilent
Anti-pancytokeratin (Cat No NB120-11213) (mc)	1/100	Novus Biologicals

* Location of manufacturers: Dako, Glostrup, Denmark; R&D systems, Minneapolis, MN, USA; Cell Signaling Technology, Danvers, MA, USA; Abcam, Cambridge, UK; Sigma-Aldrich, Saint Louis, MO, USA; Agilent, Santa Clara, CA, USA; Novus Biologicals, Colorado, USA. mc, monoclonal.

## Data Availability

Data are available from the authors upon request.

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
