# Peer review of "Macrophage-Myofibroblast Transition Contributes to Myofibroblast Formation in Proliferative Vitreoretinal Disorders"

_ijms, 2023, doi:10.3390/ijms241713510_

Round 1
Reviewer 1 Report (New Reviewer)
An interesting manuscript, but it is poorly presented. I cannot quite figure out what the authors did, why they did it and how they came to those conclusions. These are structural issues with the manuscript itself, which suggests issue with experimental design.
I did not proof the abstract as I had too many issues withthe rest of the manuscript.
Introduction: three sections.
Section 1: introduce the topic and the terminology. The first sentence is incorrect. Epiretinal membrane development is NOT a feature of proliferative vitreoretinal disorders. Instead epiretinal membrane development is "disrupted by numerous" proliferative vitreoretinal disorders such as.... This provides the connection between the region and the pathology.
section two: review pertinent literature.
line 42: two 'subtypes' of macrophages - M1 and M2.
M1 - CD86, M2 - CD206, CD163 and TGF-beta. Intermediate cell CD68. (please put that into one paragraph).
lines 70-71: intermediate between what? M1 and M2 or macrophage and myofibroblast?
Since this is a manuscript of immunohistochemistry 0 you also need to explain in the body, what is known and FAP-alpha and FSP-1 in this review section, and how they fit into all this.
section 3: goals. this is the proposal, no references in here, this is what you did.
Put numbers before each goal. e.g.: 1) to define.......2) investigate and 3) provide....
I have no idea what the authors mean by "critical attitude"? - line 77
Material and methods (skipping to the back here). Missing details here and an explanation.
4.1.: how big were the fluid samples? (line 435)
line 442 (Epiretinal...) is the start of a new paragraph.
4.2.: line 452 report section - who were they obtained? Were these obtained through paraffin histology? Cryohistology? Preservative? Technique? Type of microtome? Dewaxing protocol?
Missing chemicals not explained in the introduction include iNOS, catenin and pancytokeratin.
4.3. : Has me confused. I though this was a manuscript about the epiretinal membrane of PDR and PVR patients, in an attempt to determine the role of macrophages in these disorders. Why are the authors now culturing cells (healthy cells, I presume) of Muller cells ad HRMECs? So now authors are using two different techniques: immunohistochemistry of pathological epiretinal membranes and vitreous humour AND cell cultures of healthy retinal regions? Not explained in the introduction.
4.6.: I do not understand the fascination with box and whisker plots - average and standard deviations are more common and thus can be compared across data sets.
Results: difficult to follow. Much of the text can be converted into tables to easily compare and contrast the findings.
2.1 and 2.2: are all spindle-shaped cells myofibroblast?
2.3: what is the difference between CD68 and CD68+? They are seemingly used interchangeably.
line 131 is an interpretation (looking for M2) - stick to observations only.
sections 2.1-2.8ish can be reduced into a table with a brief mention of identity of spindle shaped cells and location of pigments (also - is this new or is this in the literature - introduction). This will get rid of much of the confusing text.
A table would easily highlight your data and show what is missing.
Example of a table (excuse the empty columns at the - technical issues)
|
Key: E = endothelium, M/M = monocytes/macrophages, MF = myofibroblasts, SSC – spindle shaped cells |
Key: E = endothelium, M/M = monocytes/macrophages, MF = myofibroblasts, SSC – spindle shaped cells
This I got from the results - note what data is missing! Also, some interpretative statements in results (e.g.: lines 148-151, 179, 200-201) - those ALL belong in the discussion.
2.9 - do NOT refer to previous studies in results - that belongs in introductions.
2.10 - this section comes out of nowhere. Yes, it is noted in the methodology section, but it is never introduced as a technique, explained why it (the technique ) was used and its significance/previous work.
Discussion: I did not read this section too carefully, as I was so confused in the results.
lines 330-334 - how does this reflect previous studies?
line 335 ( the bit in yellow) is a new paragraph. How does this reflect upon previous studied?
line 339- start new paragraph. this does explain previous studies
line 338: replace "suggest" with "indicates"
lines 370-375 come out of no where! None of these ligands were examine din this study, so why is it here. Either remove or put into context with the study.
Conclusion: this is where you bring it all together, explain what you did, what it means and what are the next steps.
First sentence need to be worded better.
The last sentence is to be deleted - as it is citing previous studies. Conclusion needs to be about future studies - not mentioned.
Figures and legends: legends are too long. I do not understand why the new highlighted parts were inserted ( about the patients - it is irrelevant).
legends need to be more uniform in construction. Either cite (A), (B), (C) at start of all descriptors or at the end of the descriptors.
Figures can be smaller.
I do not understand to what the authors are pointing in figure 6 - looks like a white line!
Figure 5D - no idea what that is. a bar graph with four equidistant dots? Just use a bar graph with the STDEV bar!
Mostly fine, just some odd word choices.
Author Response
Please see the attachment

Reviewer 2 Report (New Reviewer)
Abu El-Asrar et al. realized a very interesting article describing the “Macrophage-myofibroblast transition contributes to myofibroblast formation in proliferative vitreoretinal disorders”. Overall, this manuscript describes some potentially interesting findings on the role of macrophage-to-myofibroblast transition in epiretinal membranes from patients with proliferative diabetic retinopathy and proliferative vitreoretinopathy. The authors utilize immunohistochemistry to show co-expression of macrophage and myofibroblast markers in these tissues, suggesting an intermediate cell type undergoing transition. They also measure relevant soluble markers in vitreous samples. Additional in vitro work explores regulation of one of these markers in retinal cells.
The topic is relevant, and the results add incremental knowledge to the field. However, there are some limitations:
- The immunohistochemistry relies on co-staining of markers to identify "intermediate" cell types, but there is no definitive evidence these cells are actually transitioning macrophages. Co-expression alone does not prove transition.
- Sample sizes are quite small for the human epiretinal membrane staining (10 PDR, 11 PVR patients). Larger sample sizes would strengthen conclusions.
- For the vitreous biomarkers, inclusion of a non-proliferative diabetic retinopathy group could help determine if levels correlate with proliferation severity.
- The in vitro work on FAP-alpha regulation, while interesting, seems somewhat disconnected from the rest of the study. The relevance to the human disease context is unclear.
- The writing could be tightened in places - there is some redundancy between results and discussion.
Here are some suggestions for revisions that could help strengthen the manuscript:
Introduction:
- Provide more background on the different macrophage activation states (M1 vs M2), and evidence that M2 is pro-fibrotic while M1 is anti-fibrotic. This will help set up the rationale for studying M2 markers.
- Expand on prior evidence of macrophage-to-myofibroblast transition in other diseases/organs. This will strengthen the rationale for studying this phenomenon in the eye.
Methods:
- Increase sample sizes for human epiretinal membranes if possible, to have more confidence in generalizing the results.
- For vitreous biomarkers, include a non-proliferative diabetic retinopathy group as an additional comparison to determine if levels correlate with proliferation severity.
- Provide details on how the in vitro work relates to the human disease context. For example, are the concentrations of TNF-α and VEGF physiologically relevant? How do the cultured cells relate to cells in epiretinal membranes?
Results:
- For immunostaining, include negative controls and quantification (e.g. % double positive cells) in addition to the qualitative descriptions. This will strengthen the characterization of double positive transitional cells.
- Streamline descriptions of immunostaining results to avoid redundancy between figure legends and main text.
Discussion:
- Shorten the first paragraph reiterating the results. Focus more on interpretation and limitations.
- Discuss alternative explanations for the double positive cells besides transitional states (e.g. separate cells in close proximity).
- Discuss limitations of using marker co-expression to infer transition. Suggest future experiments (in vitro induction, fate mapping) that could more definitively demonstrate transition.
- Relate the in vitro findings back to the human disease context. Are the concentrations of TNF-α etc physiologically relevant? How do the cultured cells relate to cells in epiretinal membranes?
· Furthermore, I suggest adding data related to recent bulk transcriptomics studies which could represent a strong substrate to enforce the role of described molecular mechanisms, such as the recent PMID: 36290689, PMID: 36490268 and PMID: 32184807.
In summary, providing more context, details, controls, and cautious interpretation could help strengthen the manuscript and account for limitations. The topic is interesting, but more work is needed to conclusively demonstrate macrophage-to-myofibroblast transition.
The English language need important revisions.
Author Response
Please see the attachment.

This manuscript is a resubmission of an earlier submission. The following is a list of the peer review reports and author responses from that submission.
Round 1
Reviewer 1 Report
Abu El-Asrar. Et. al. investigated the role of M2 macrophages in fibrosis development in proliferative vitreoretinal disorders. The authors identify macrophage-myofibroblast transition (MMT) mainly based on their IHC observations. Some potential therapeutic strategies were suggested for preventing the MMT driven disease progression. The manuscript should be further improved by experimentally addressing the following concerns:
Major
1. Source of myofibroblasts: It is important to validate the source of myofibroblasts as being derived from macrophages. A depletion study (PMID: 30054470, 36206343) where macrophages are selectively removed or reduced can help confirm this hypothesis. This would strengthen the claim made about macrophage-myofibroblast transition (MMT).
2. M2 marker: CD206 is commonly used as an M2 marker, but its expression is not exclusive to M2 macrophages. Macrophage specific markers such as human CD68, mouse F4/80 should be co-stained with the CD206 for identifying the M2 cells.
3. TGF-beta/Smad3 signalling pathway is reported as a major pathway for MMT formation. However, the involvement of beta-catenin requires further validation or discussion.
4. The functional impact of MMT in vivo could be evaluated more comprehensively with adoptive transfer studies (PMID: 34791825, 22745325). The causation effect of MMT on disease progression should be demonstrated experimentally.
5. Quantification: To further substantiate the findings, consider using immunofluorescence (IF) and flow cytometry for quantification of double-staining experiments. These methods would provide more precise, quantitative data to support the observations.
6. In vitro model: Developing an in vitro model where macrophages are stimulated under disease-mimicking conditions could provide more detailed insights into the mechanisms of MMT. This approach could help elucidate the triggers for this transition in a controlled setting.
Minor
Regarding lines 149, 309, and 313: All data that are mentioned in the results section should be clearly shown in the manuscript. If the data are not directly included in the main text, they should be included as supplemental figures or tables, with clear references in the text.
Author Response
Dear Reviewers,
The following is the rational of our study:
In this study, we investigated the role of macrophage-myofibroblast transition (MMT) in the contribution to myofibroblast population present in epiretinal membranes from patients with proliferative vitreoretinal disorders. We examined 10 membranes from patients with proliferative diabetic retinopathy (PDR) obtained during pars plana vitrectomy for the repair of tractional retinal detachment and 11 membranes obtained from patients undergoing vitreoretinal surgery for the treatment of retinal detachment complicated by proliferative vitreoretinopathy (PVR).
- In previous studies, the demonstration of MMT in human diseases relies on the detection of intermediate cells that co-express macrophage markers, such as CD68 and a myofibroblast marker, such as α-smooth muscle actin (α-SMA). (References 12-19)
- In the present study, we used the fibroblast/myofibroblast markers α-SMA, fibroblast activation protein-α (FAP-α) and fibroblast-specific protein-1 (FSP-1) and we compared PDR with PVR (with control samples from non-diabetic patients).
- We demonstrated that most of the CD68+ monocytes/macrophages had the M2 phenotype, identified by co-expression of the M2 macrophage marker CD206, but not the M1 macrophage markers, CD86 and iNOS.
- In this study, we demonstrated the existence of MMT in epiretinal membranes from patients with PDR and PVR by identifying the presence of intermediate cells co-expressing CD68 and the fibroblast/myofibroblast markers α-SMA and FSP-1. In addition, we demonstrated that CD206+ M2 macrophages co-expressed α-SMA, FSP-1 and FAP-α.
- In addition, we demonstrated the expression of TGF-ß1/ß-catenin signaling pathway in PDR and PVR epiretinal membranes.
- FAP-α is a marker of myofibroblast activation and the original membrane-bound form can be shed as an active soluble form (sFAP-α). FAP-α is involved in the pathogenesis of angiogenic, inflammatory and fibrotic disorders. CD206 is also shed from activated macrophages to generate (a) soluble form(s) (sCD206). We, therefore, analyzed vitreous fluid samples from 38 patients with PDR, 10 patients with PVR and 30 nondiabetic controls for the presence of these biomarkers. We demonstrated significant upregulation of these biomarkers in the vitreous fluid from patients with proliferative vitreoretinal disorders. There was a significant positive correlation between their levels.
- As retinal microvascular endothelial cells and retinal Müller glial cells are major cell types in inflammatory reactions in the retina, we investigated the effect of various proliferative vitreoretinal disorders-associated mechanisms on the expression of sFAP-α in cultured human retinal microvascular endothelial cells and human retinal Müller glial cells to identify additional cellular sources for sFAP-α. We demonstrated the capability of the proinflammatory cytokine TNF-α and hypoxia to induce production of sFAP-α in Müller cells and that inhibition of the inflammatory transcription factor NF-Ä¸ß significantly attenuated TNF-α-induced upregulation of sFAP-α in Müller cells.
Answers to Reviewer#1
The authors thank the reviewer for his/her comments and the time spent on the revision.
Abu El-Asrar. et al. investigated the role of M2 macrophages in fibrosis development in proliferative vitreoretinal disorders. The authors identify macrophage-myofibroblast transition (MMT) mainly based on their IHC observations. Some potential therapeutic strategies were suggested for preventing the MMT-driven disease progression. The manuscript should be further improved by experimentally addressing the following concerns:
Major
- Source of myofibroblasts: It is important to validate the source of myofibroblasts as being derived from macrophages. A depletion study (PMID: 30054470, 36206343) where macrophages are selectively removed or reduced can help confirm this hypothesis. This would strengthen the claim made about macrophage-myofibroblast transition (MMT).
Answer: We agree that depletion studies help in preclinical experiments to some degree to pinpoint specific cell types. We are, however, dealing here with real human samples from surgical specimens and not with an animal model allowing depletion studies.
- M2 marker: CD206 is commonly used as an M2 marker, but its expression is not exclusive to M2 macrophages. Macrophage-specific markers such as human CD68, mouse F4/80 should be co-stained with the CD206 for identifying the M2 cells.
Answer: We agree with this point. To the best of our possibilities, we have addressed this point, keeping in mind the differences between human (our study) and mouse studies (F4/80 is a typical macrophage marker in mice, but not in human). We provided CD68 staining (Figure 3) and CD68/CD206 co-staining (Figure 4). Our findings were corroborated by 3 recent studies (references 11, 27, 28), Discussion, page 18, line 379.
- TGF-beta/Smad3 signalling pathway is reported as a major pathway for MMT formation. However, the involvement of beta-catenin requires further validation or discussion.
Answer: We have provided experimental evidence by showing expression of TGF-ß1 and ß-catenin and ß-catenin/CD206 co-expression (Figures 10, 11, 12). More details are added to the Discussion (page 19, line 403).
- The functional impact of MMT in vivo could be evaluated more comprehensively with adoptive transfer studies (PMID:34791825, 22745325). The causation effect of MMT on disease progression should be demonstrated experimentally.
Answer: We agree that with preclinical studies (as indicated with two references) one may provide some evidence about causal relations. This would, however, constitute a completely different study with pitfalls that may even not be resolved on long terms (differences between species, adequate animal models for PDR and PVR, etc). We have added these statements and used thankfully one of the two indicated references (Reference 19). Please consider that we are studying human material obtained during surgery and that in all previous studies dealing with the process of MMT, the investigators depended on the co-staining of CD68 and α-SMA.
- Quantification: To further substantiate the findings, consider using immunofluorescence (IF) and flow cytometry for quantification of double-staining experiments. These methods would provide more precise, quantitative data to support the observations.
Answer: We agree that quantification is important is research and we have done our best to include quantitative data where possible. The surgically obtained human epiretinal membranes are small and do not yield enough cells to perform meaningful flow cytometry analysis. To the best of our knowledge, such analysis has not been done so far on the membranes, though sometimes vitreous is subjected to flow cytometry.
- In our laboratory, we have not tested and validated double immunofluorescence protocols on our 2-µm thick paraffin-embedded tissue sections for several reasons:
o The cutting of the paraffin blocks and the correct mounting of the tissues on the glass slides (i.e. without tearing or causing wrinkles in these thin sections) is a difficult procedure, for which we want to obtain an optimal result before starting with the next steps of the technical process.
o We have had the experience that the common methods of tissue pre-treatment for unmasking antigens, such as boiling for variably long periods in buffered solutions of chemicals at different pHs, or enzymatic digestion, may be too corrosive for small pieces of tissue which are moreover very thinly cut. This may have an impact on the detectability of those antigens that are present at low-levels or that are difficult to stain due to their physicochemical characteristics.
o We may predict that no single, or even multiple, pre-treatment methods would work equally well for all the different antigens that we want to (double-)stain as their locations, concentrations, physical accessibility and chemical stainability are often a priori unknown. An immunofluorescence (or double-immunofluorescence) approach might be impossible for some of the studied molecules.
o We feel that the problems of the determination of the cellular colocalization of specific antigens, and the identification of the cell types or tissue components in which they are present, would be amplified by the dark-field examination necessary for double-immunofluorescence techniques. For correct cellular identification, we might have to switch all the time between dark-field double-immunofluorescence and bright-field white light microscopy on corresponding haematoxylin-eosin stained slides.
o Photographic documentation of multiple targets would cost more time and might be less accurate or at least more subject to discussion than with bright-field double immunohistochemistry.
o Lastly, with the use of the immunofluorescence technique the problem of photobleaching or fading exists that is mostly absent with immunohistochemistry. As we store most research slides in our lab for a very long time, immunohistochemistry seems the most appropriate technique.
- We of course agree with the reviewer that immunofluorescence has certain applications for which it is an optimal investigational technique. This can be on:
o fresh-frozen tissues (we have no epiretinal membrane tissues in this condition available in our lab);
o formalin-fixed paraffin-embedded tissues with abundantly available, easily stainable and chemically stable antigens;
o cell cultures.
- In vitro model: Developing an in vitro model where macrophages are stimulated under disease-mimicking conditions could provide more detailed insights into the mechanisms of MMT. This approach could help elucidate the triggers for this transition in a controlled setting.
Answer: We agree that in vitro experiments could help in dissecting the triggers, different steps and kinetics of the process in vitro. As this would require more time than granted for revision, this will be subject of a follow-up study. The proposed experiment has somehow been performed by others and those studies have been discussed (page 18; line 404; References 12, 17). We have revealed TGF-ß1 and ß-catenin expression and ß-catenin/CD206 co-staining (Figures 10, 11, 12). This is an important pathway that is known to be involved in fibrosis in general, but apparently also in the MMT process in PDR and PVR.
Minor
Regarding lines 149, 309, and 313: All data that are mentioned in the results section should be clearly shown in the manuscript. If the data are not directly included in the main text, they should be included as supplemental figures or tables, with clear references in the text.
Answer: We have now included all these data (Figure 5D and in a table; Page 16). Thank you.

Reviewer 2 Report
The proposed manuscript deals with the highly relevant topic of the origin of myofibroblasts at the vitreoretinal interface in patients with proliferative vitreoretinal disease. The used techniques are sound, however, further experiments and more precise discussion of the shown results is needed, in order to publish the data in a high-impact journal.
p. 2, ll. 68-70: Why do the authors assume that the immune cells shown in this study are monocytes, i.e. infiltrating cells? Isn't the more obvious option that these are tissue-specific resident myleloid cells (hyalocytes or retinal microglia)? This should at least be discussed later on in the Discussion and mentioned in the Introduction.
p. 2, l. 73-74: "To our knowledge, currently no studies exist about MMT in proliferative vitreoretinal disorders."
There are actually two studies that pretty much cover the topic of the proposed manuscript: on the origin of immune cells in epiretinal proliferative vitreoretinopathy membranes (Laich et al., IOVS 2022 - PMID: 35579905), as well as in retinal neovacularizations from PDR patients (Boneva et al., Front Immunol 2021 - PMID: 34795670) – in these cases, by state-of-the-art techniques, such as imaging mass cytometry and RNA sequencing. One of the studies is briefly mentioned in the Discussion. Unfortunately, the existence of these two studies reduces the novelty character of the current study.
p. 18, ll. 419-425: The authors should define their patients’ cohort better. Was the surgery of the PVR patients a primary one? The presence and respectively absence (in patients with a secondary retinal detachment after first vitrectomy for retinal detachment) of vitreous tissue in the sample material would surely influence the following cytokine analysis. Further, were diabetes mellitus or diabetic retinopathy excluded in control and PVR patients?
The authors should consider a control group not only for the cytokine analyses, but also for membrane tissue. Were all samples stained for all markers or only a certain number out of 10, resp. 11 membranes? These informations should be included in the Methods section.
Figure 1E.: This reviewer does not find the description of this staining in the Results section very convincing: There are no “new” blood vessels lined with FSP-1-positive cells (p.2, ll. 89-91) shown in this extract. Simultaneously, these are not even mentioned in the legend of Figure 1E.
Figure 2D: Most of the cells tagged by arrows are not spindle-shaped.
Figure 3B and section 2.5: In fact, the most interesting and actually novel finding in this study is the description of pigment IN immune cells at sites of PVR membranes, in contrast to pigment epithelial cells. This should be highlighted and discussed later, since it is in contrast with the current dogma of the origin of myofibroblasts in PVR membranes.
Figure 4: Please highlight some single-positive cells, since this staining is not quite convincing for this reviewer – as I see it, in the PDR samples, the CD206 marker is very weak (however, quite specific), while in the PVR sample this reviewer can barely distinguish red from brown staining. At the same time, the assumption of pigment-containing cells in PVR membranes, would make it very difficult for a mindful reader to distinguish CD68-positive cells from pigment-containing cells. The same is valid for Figure 7C and D and Figure 9.
Figure 6B and C: Although, as stated above, this is the most interesting finding of this study, this reviewer does not find the stainings of serial sections quite convincing. The authors should consider co-stainings. Are these really serial sections?
p. 6, ll. 146-148: Why do the authors assume that the CD206-positive cells in the vicinity of blood vessels are of non-myeloid origin? Perivascular macrophages are well-known to be positive of CD206. In order to state that these are endothelial cells, also positive for CD206, the authors should be delivering a CD31 or an alternative endothelial co-staining. Alternatively, these could be antigen-presenting endothelial cells. This reviewer would leave this open, since the approach is not sufficient to tell these cell populations apart.
p. 7, ll. 165-166: This is a false statement. In 5B there is no co-staining for α-SMA, meaning that while the cells are spindle-shaped and most probably myfibroblasts, this cannot be stated for sure.
The authors should consider showing data of absent M1-specific staining in PDR and PVR membranes in a supplementary figure.
p. 12, ll. 239-240: TGF-β1-positive cells should rather be described as cells lining blood vessels than called “vascular endothelial cells”. The same is valid for p. 12, l. 245 and p. 13, ll. 258-259.
p. 12, l. 252: The cells tagged by arrows in Figure 11B are not really “spindle-shaped”…
Why was a co-staining only performed for β-catenin and CD206 and not for TGF-β1 and CD206?
pp. 16-17, ll. 329-342: For this reviewer, the discussion in this paragraph is not quite conclusive. Is FAP-α-positivity only characteristic for smooth muscle cells and transitioning cells (in this case M2) and NOT for myeloid cells? What is the explanation for CD68-positive cells, also positive for α-SMA? The authors should elucidate on this matter in more detail.
p. 17, ll. 342-346: The authors should mention that in this particular study, it has been shown by immunohistochemistry that co-expression of a myeloid cell marker (Iba1) with α-SMA has already been demonstrated…
p. 17, l. 358: In this study, there is no co-staining of e.g. CD31 and β-catenin, in order to claim with certainty that the vessel-lining cells shown are endothelial cells. This reviewer agrees that these, of course, most probably are endothelial cells, however, this should be phrased accordingly.
p. 17, l. 384: This a quite diffuse statement – above, the authors state that FAP-α is characteristic of MMT transition and smooth muscle cells. What about myofibroblasts of another origin? The authors should clarify their statements and support them with appropriate references.
p. 17, l. 384-385: The link between myofibroblasts, their contribution to fibrotic events at the vitreoretinal interface and secretion of FAP-α by Müller cells remains very unclear to this reviewer, even after careful reading of the Discussion.
p. 18, ll. 390-398: As discussed in detail above, there are other possibilities for the origin of CD206-positive endothelial cells and these should at least (!) be discussed.
Discussion: This section as a whole is very diffuse and chaotic, and lacking a golden thread.
The quality of English language in this manuscript is sufficient and only requires minor editing.
Author Response
Dear Reviewers,
The following is the rational of our study:
In this study, we investigated the role of macrophage-myofibroblast transition (MMT) in the contribution to myofibroblast population present in epiretinal membranes from patients with proliferative vitreoretinal disorders. We examined 10 membranes from patients with proliferative diabetic retinopathy (PDR) obtained during pars plana vitrectomy for the repair of tractional retinal detachment and 11 membranes obtained from patients undergoing vitreoretinal surgery for the treatment of retinal detachment complicated by proliferative vitreoretinopathy (PVR).
- In previous studies, the demonstration of MMT in human diseases relies on the detection of intermediate cells that co-express macrophage markers, such as CD68 and a myofibroblast marker, such as α-smooth muscle actin (α-SMA). (References 12-19)
- In the present study, we used the fibroblast/myofibroblast markers α-SMA, fibroblast activation protein-α (FAP-α) and fibroblast-specific protein-1 (FSP-1) and we compared PDR with PVR (with control samples from non-diabetic patients).
- We demonstrated that most of the CD68+ monocytes/macrophages had the M2 phenotype, identified by co-expression of the M2 macrophage marker CD206, but not the M1 macrophage markers, CD86 and iNOS.
- In this study, we demonstrated the existence of MMT in epiretinal membranes from patients with PDR and PVR by identifying the presence of intermediate cells co-expressing CD68 and the fibroblast/myofibroblast markers α-SMA and FSP-1. In addition, we demonstrated that CD206+ M2 macrophages co-expressed α-SMA, FSP-1 and FAP-α.
- In addition, we demonstrated the expression of TGF-ß1/ß-catenin signaling pathway in PDR and PVR epiretinal membranes.
- FAP-α is a marker of myofibroblast activation and the original membrane-bound form can be shed as an active soluble form (sFAP-α). FAP-α is involved in the pathogenesis of angiogenic, inflammatory and fibrotic disorders. CD206 is also shed from activated macrophages to generate (a) soluble form(s) (sCD206). We, therefore, analyzed vitreous fluid samples from 38 patients with PDR, 10 patients with PVR and 30 nondiabetic controls for the presence of these biomarkers. We demonstrated significant upregulation of these biomarkers in the vitreous fluid from patients with proliferative vitreoretinal disorders. There was a significant positive correlation between their levels.
- As retinal microvascular endothelial cells and retinal Müller glial cells are major cell types in inflammatory reactions in the retina, we investigated the effect of various proliferative vitreoretinal disorders-associated mechanisms on the expression of sFAP-α in cultured human retinal microvascular endothelial cells and human retinal Müller glial cells to identify additional cellular sources for sFAP-α. We demonstrated the capability of the proinflammatory cytokine TNF-α and hypoxia to induce production of sFAP-α in Müller cells and that inhibition of the inflammatory transcription factor NF-Ä¸ß significantly attenuated TNF-α-induced upregulation of sFAP-α in Müller cells.
Answers to Reviewer#2
The authors thank the reviewer, who asks for minor edition and to improve the cited articles, the study design, presentation and conclusion
Specific questions
The proposed manuscript deals with the highly relevant topic of the origin of myofibroblasts at the vitreoretinal interface in patients with proliferative vitreoretinal disease. The used techniques are sound, however, further experiments and more precise discussion of the shown results is needed, in order to publish the data in a high-impact journal.
Point 1.
- 2, ll. 68-70: Why do the authors assume that the immune cells shown in this study are monocytes, i.e. infiltrating cells? Isn't the more obvious option that these are tissue-specific resident myeloid cells (hyalocytes or retinal microglia)? This should at least be discussed later on in the Discussion and mentioned in the Introduction.
Answer: We thank the reviewer for insisting to detail this. When we wrote about monocytes/macrophages we did not want to stress details about their origins and we implied both hematogenic and local cell origins. We thought that this aspect would be beyond the details of our study, and the CD68 marker used cannot discriminate between different cells (retinal microglia, infiltrating monocytes, in situ proliferating macrophages). Hyalocytes do not express CD68. On the request of the reviewer, have now added to the Introduction and Discussion details about these aspects (Introduction, page 2; line 66; Discussion, page 18; line 379).
Point 2.
- 2, l. 73-74: "To our knowledge, currently no studies exist about MMT in proliferative vitreoretinal disorders."
There are actually two studies that pretty much cover the topic of the proposed manuscript: on the origin of immune cells in epiretinal proliferative vitreoretinopathy membranes (Laich etal., IOVS 2022 - PMID: 35579905), as well as in retinal neovacularizations from PDR patients (Boneva et al., Front Immunol 2021 - PMID: 34795670) – in these cases, by state-of-the-art techniques, such as imaging mass cytometry and RNA sequencing. One of the studies is briefly mentioned in the Discussion. Unfortunately, the existence of these two studies reduces the novelty character of the current study.
Answer: We apologize for having overseen one of those references. We have deleted our sentence in the Introduction. We respectfully disagree with the claim that our study is undermined in novelty by the two mentioned studies. First of all, we did not briefly mention the study by Laich et al. but placed this study carefully in relation to our work. Because both indicated studies came from the same excellent Freiburg group of researchers, we relied for the most up-to date information on their most recent study from 2022 (including Dr. Boneva and colleagues), rather than on the study by Boneva et al. from 2021.
Furthermore, Laich et al. compared samples from patients with retinal detachment due to PVR (n= 19); with idiopathic macular pucker (n= 13) and internal limiting membranes from idiopathic macular hole patients (n=7), whereas in the study by Boneva, 7 samples came from PDR patients, 10 from patients with idiopathic macular pucker and 7 with macular hole.
We based our findings on comparisons between PDR with PVR, the two most important proliferative vitreoretinal disorders and these comparisons were not yet done. In addition, the new techniques of mass cytometry and RNA sequencing are indeed modern and state-of-the-art. On the bulk RNAseq data, the authors used software to post-sequencing discriminate between different cell types. This analysis suggested that M2 macrophages are present, which was confirmed by the authors using classical IHC. However, our study mainly deals with the comparison between PDR and PVR. In this way, our data are revealing new information that is not repetitive but instead complementary. For instance, we used other markers for myofibroblasts (FSP-1 and FAP-α) to better delineate the sequence of MMT. We agree that the study by Boneva and colleagues (including Dr Yannick Laich) may be mentioned too in our study because (i) it predates the study by Laich et al. and (ii) it discusses a series of samples from different patient cohorts. We thank the reviewer for helping us to improve our study in this way and we have now also added and discussed the reference by Boneva and colleagues (Introduction, page 2; Discussion, page 18; Reference 11).
Point 3.
- 18, ll. 419-425: The authors should define their patients’ cohort better. Was the surgery of the PVR patients a primary one? The presence and respectively absence (in patients with a secondary retinal detachment after first vitrectomy for retinal detachment) of vitreous tissue in the sample material would surely influence the following cytokine analysis. Further, were diabetes mellitus or diabetic retinopathy excluded in control and PVR patients?
The authors should consider a control group not only for the cytokine analyses, but also for membrane tissue. Were all samples stained for all markers or only a certain number out of 10, resp. 11 membranes? These informations should be included in the Methods section.
Answer: On the request of the reviewer, we have added now more details about the cohorts of patients/ Specifically:
- All PVR patients underwent vitreoretinal surgery for the treatment of primary rhegmatogenous retinal detachment complicated by PVR (Materials and Methods, page 21).
- Control subjects were clinically checked to be free from diabetes (Materials and Methods, page 21).
- PVR membranes were from patients without diabetes (Materials and Methods, page 21).
- All membranes were stained for all markers as indicated in the Results section and Figure Legends.
- Samples from patients with macular hole or macular pucker are not real controls as these samples are obtained from patients with other vitreoretinal interface disorders. Moreover, we believe that all preretinal membranes are pathological in nature. In addition, we do not have good quality biopsies from these patients that can provide reliable immunohistochemical staining results. Therefore, we usually compare PDR (fibrovascular) and PVR (fibrocellular) membranes.
Point 4.
Figure 1E.: This reviewer does not find the description of this staining in the Results section very convincing: There are no “new” blood vessels lined with FSP-1-positive cells (p.2, ll. 89-91) shown in this extract. Simultaneously, these are not even mentioned in the legend of Figure 1E.
Answer: We added an extra panel to show FSP-1 staining in endothelial cells lining pathologic new blood vessels (Results, page 3; Figure 1E; Legends).
Point 5
Figure 2D: Most of the cells tagged by arrows are not spindle-shaped.
Answer: We have done our best to indicate spindle-shaped cells in the new version. A new Figure 2D is added.
Point 6
Figure 3B and section 2.5: In fact, the most interesting and actually novel finding in this study is the description of pigment IN immune cells at sites of PVR membranes, in contrast to pigment epithelial cells. This should be highlighted and discussed later, since it is in contrast with the current dogma of the origin of myofibroblasts in PVR membranes.
Answer: As stated above about in situ demonstration and comparisons of MMT in human PDR and PVR eye tissues as an important finding complementing other studies, we thank the reviewer for pointing to another interesting finding by our observations, namely the phagocytosis of pigment. We do not want to exaggerate this finding and the fact that we would reverse a dogma, but rather want to convince the readership about alternative paradigms (rather than dogmas). We agree that both MMT and pigment phagocytosis, represent elements that make our study worthwhile to publish. As requested, we discussed this finding (Discussion, page 18, line 360).
Point 7
Figure 4: Please highlight some single-positive cells, since this staining is not quite convincing for this reviewer – as I see it, in the PDR samples, the CD206 marker is very weak (however, quite specific), while in the PVR sample this reviewer can barely distinguish red from brown staining. At the same time, the assumption of pigment-containing cells in PVR membranes, would make it very difficult for a mindful reader to distinguish CD68-positive cells from pigment-containing cells. The same is valid for Figure 7C and D and Figure 9.
Answer. At the request of the reviewer, we have now indicated single stained cells that were positive only for CD206 (red) with the use of arrowheads in Figure 4. Please note that these are all spindle shaped cells. As a further point of discussion, the brown staining of CD68 should not be a problem in PDR samples (because in these patients there is no such problem with the confusion between CD68 and brown pigment). All immunohistochemical evaluations were done by a very experienced pathologist (Prof. G. De Hertogh). The dark brown (almost black) color of the pigment is also different from the brown of the DAB pigment. (Figure 4: Legend).
Point 8
Figure 6B and C: Although, as stated above, this is the most interesting finding of this study, this reviewer does not find the stainings of serial sections quite convincing. The authors should consider co-stainings. Are these really serial sections?
Answer: We thank the reviewer for suggesting that she/he evaluates/appreciates this as the most interesting finding. By the fact that we placed emphasis on MMT and on comparison between PDR and PVR, we think that our observations help with these matters as such. Indeed, Figure 6 shows serial sections (in slightly different orientations). To make this more clear, we drew a line around the cell group of interest in the revised figure (Figure 6). By this indication it should become easier to recognize that these are serial sections.
Point 9
- 6, ll. 146-148: Why do the authors assume that the CD206-positive cells in the vicinity of blood vessels are of non-myeloid origin? Perivascular macrophages are well-known to be positive of CD206. In order to state that these are endothelial cells, also positive for CD206, the authors should be delivering a CD31 or an alternative endothelial co-staining. Alternatively, these could be antigen-presenting endothelial cells. This reviewer would leave this open, since the approach is not sufficient to tell these cell populations apart.
Answer:
- To avoid confusion, we changed the title of the paragraph to “CD206+ cells in epiretinal membranes from patients with PDR and PVR (Results, page 7).
- CD206 immunoreactivity was observed in endothelial cells lining pathologic new blood vessels in PDR membranes (Figure 5A). We performed staining with the endothelial cell marker CD31 (Figure 1B) and indeed the cells lining the vessels have a similar phenotype as the CD31 staining. In addition, we confirmed expression of CD206 in cultured endothelial cells by flow cytometry. We, therefore, prefer to keep the endothelial cell indication. In addition, CD206 immunoreactivity was observed in stromal monocytes/macrophages (Figure 5B) and stromal spindle-shaped cells (Figure 5C). In PVR membranes, CD206 immunoreactivity was observed in monocytes/macrophages (Figure 5D) and spindle-shaped cells (Figure 5E).
Point 10
- 7, ll. 165-166: This is a false statement. In 5B there is no co-staining for α-SMA, meaning that while the cells are spindle-shaped and most probably myfibroblasts, this cannot be stated for sure.
Answer: We do not understand what the reviewer means with ”false statement”, as we did not use the word myofibroblasts here, but only indicated the immune reactive cells to be spindle-shaped in the legend of Figure 5C.
Point 11
The authors should consider showing data of absent M1-specific staining in PDR and PVR membranes in a supplementary figure.
Answer: It is almost impossible to show absence of any cell type, as such concepts rely on detection limits, sample sites etc. Like with most approaches, our studies are based on the presence of positive detection signals. We performed the staining for the M1 markers as indicated in the manuscript (Materials and Methods, page 21, 22). We did not detect M1 immunoreactivity (iNOS, CD86) in the PDR or PVR membranes.
Point 12
- 12, ll. 239-240: TGF-β1-positive cells should rather be described as cells lining blood vessels than called “vascular endothelial cells”. The same is valid for p. 12, l. 245 and p. 13, ll.258-259.
Answer: See above. We indicated endothelial cells with typical morphology as shown by CD31 (endothelial cell marker) staining (Figure 1B). We have complied with this request at the indicated places, thank you.
Point 13
- 12, l. 252: The cells tagged by arrows in Figure 11B are not really “spindle-shaped”…
Why was a co-staining only performed for β-catenin and CD206 and not for TGF-β1 and CD206?
Answer: We have deleted some arrows to convince the reviewer and only kept arrows indicating spindle-shaped cells. The reason why we did not do co-staining between TGF-ß1 and CD206 was because the cells expressing TGF-ß1 displayed mostly a spindle-shaped morphology and did not have a monocyte/macrophage phenotype.
Point 14
- 16-17, ll. 329-342: For this reviewer, the discussion in this paragraph is not quite conclusive. Is FAP-α-positivity only characteristic for smooth muscle cells and transitioning cells (in this case M2) and NOT for myeloid cells? What is the explanation for CD68-positive cells, also positive for α-SMA? The authors should elucidate on this matter in more detail.
Answer: In accordance with the question raised by the reviewer, we have elaborated on this matter in the Discussion section as follows. “The finding of specific cell markers in transitioning cell types indicates plasticity of cell phenotypes. The importance of such findings is in the understanding of biological processes and eventually in mastering such processes to the benefit of patients. Before this may happen, more details will be needed and the final test will always be with convincing (pre)clinical studies.”
We have prepared a scheme (new Figure 15) that illustrates the transition from monocyte to myofibroblast in PDR and the appearance/disappearance of markers, based on our observations. The CD68-positive cells also positive for α-SMA are transitioning cells in an intermediate state as indicated (Discussion, page 19).
Point 15
- 17, ll. 342-346: The authors should mention that in this particular study, it has been shown by immunohistochemistry that co-expression of a myeloid cell marker (Iba1) with α-SMA has already been demonstrated…
Answer: We added this information to the discussion (Discussion, page 18, line 379)
Point 16
- 17, l. 358: In this study, there is no co-staining of e.g. CD31and β-catenin, in order to claim with certainty that the vessel-lining cells shown are endothelial cells. This reviewer agrees that these, of course, most probably are endothelial cells, however, this should be phrased accordingly.
Answer: See answer to point 9. We always perform CD31 staining in PDR membranes. ß-catenin staining in cells lining the pathologic new blood vessels (Figure 11B) is similar to CD31 staining (Figure 1B).
Point 17
- 17, l. 384: This a quite diffuse statement – above, the authors state that FAP-α is characteristic of MMT transition and smooth muscle cells. What about myofibroblasts of another origin? The authors should clarify their statements and support them with appropriate references.
Answer: FAP-α is an alternative marker for myofibroblast (Introduction, page 2, last paragraph). In addition, FAP-α marks myofibroblast activation (Discussion, page 20, line 422).
Point 18
- 17, l. 384-385: The link between myofibroblasts, their contribution to fibrotic events at the vitreoretinal interface and secretion of FAP-α by Müller cells remains very unclear to this reviewer, even after careful reading of the Discussion.
Answer. To enhance the findings of the manuscript we wanted to measure by ELISA in vitreous samples sCD206 and sFAP-α to confirm the immunohistochemical detection of those markers in the membranes. The levels of sCD206 and sFAP-α were upregulated in both types of vitreous samples. Finally, we checked production of the soluble markers by cell types known to contribute to the inflammation in the eye, namely endothelial cells and Müller cells and demonstrated that Müller cells were more efficient in producing sFAP-α compared to human retinal microvascular endothelial cells. The production of sFAP-α was downstream of the NF-ĸB inflammatory pathway, as a specific inhibitor reduced the production.
We thereby demonstrated that specific cells, e.g. Müller cells, may contribute more to biological processes by FAP than previously recognized. Our sentence is only written to mediate that fact. (Discussion, page 20, paragraphs 2, 4).
Point 20
- 18, ll. 390-398: As discussed in detail above, there are other possibilities for the origin of CD206-positive endothelial cells and these should at least (!) be discussed.
Answer: As we agreed in point 9, we have taken this remark well and have included the possibility that perivascular macrophages express CD206.

Round 2
Reviewer 1 Report
Suprisingly, the authors did not make any efforts to response to my concerns.
Importantly, MMT derived myofibroblasts can never be detected by immunostaining with only a single marker. I cannot believe the proposed work due to the recent findings.
N/A